# Estimating hourly ground-level aerosols using GEMS aerosol optical depth: A machine learning approach

Sungmin O[1], Ji Won Yoon[2,3], and Seon Ki Park[2,3,4]

[1]Department of AI Software, Kangwon National University, Samcheock, Republic of Korea
[2]Center for Climate/Environment Change Prediction Research, Ewha Womans University, Seoul, Republic of Korea
[3]Severe Storm Research Center, Ewha Womans University, Seoul, Republic of Korea
[4]Department of Climate and Energy Systems Engineering, Ewha Womans University, Seoul, Republic of Korea

**Correspondence:** Seon Ki Park (spark@ewha.ac.kr)

**Abstract.** The Geostatellite Environment Monitoring Spectrometer (GEMS) is the world's first ultraviolet–visible instrument for air quality monitoring in geostationary orbit. Since its launch in 2020, GEMS has provided hourly daytime air quality information over Asia. However, to date, validation and applications of these data are largely lacking. Here we evaluate the effectiveness of the first two years of GEMS aerosol optical depth (AOD) data in estimating ground-level particulate matter (PM) concentrations at an hourly scale. To do so, we train random forest and XGBoost machine learning algorithms using GEMS AOD data and meteorological variables as input features, then employ the trained models to estimate PM10 and PM2.5 concentrations in South Korea. The model-estimated PM concentrations well capture the spatial and temporal variations observed in ground-based measurements, showing strong correlations. However, they exhibit noticeable biases at the extremes, with a tendency to overestimate concentrations at lower PM levels and underestimate them at higher PM levels. Incorporating locally available data, such as carbon monoxide and nitrogen dioxide measurements, into the model training further enhances performance, improving correlations and reducing errors. Moreover, we demonstrate the feasibility of using machine learning models with neighboring station data to estimate PM concentrations at ungauged locations where ground PM measurements are not available. Our results will serve as a reference to aid the evaluation of future GEMS AOD retrieval algorithm improvements and also provide initial guidance for data users.

## 1 Introduction

The adverse impacts of particulate matter (PM) on human health are well known. Exposure to high PM concentration can cause serious health risks such as cancers, respiratory and cardiovascular diseases (Chen and Hoek, 2020; Kim and Kim, 2020; Ciabattini et al., 2021; Moreno-Ríos et al., 2022). PM can also have a harmful effect on ecosystems through deposition of PM and its subsequent uptake by plants (Rai, 2016; Roy et al., 2024). Accordingly, in many countries, it is mandatory to control ambient PM concentrations, and regular PM concentration measurements are key to designing appropriate policies to constrain

the presence of PM. Given this background, the number of air quality monitoring stations has been growing worldwide; however, these ground-based measurement stations are often concentrated in city areas only and sparsely distributed to provide spatially continuous data (Martin et al., 2019).

In contrast, satellite observational data, with its broad spatial coverage, can be potentially used to improve air quality monitoring (including PM) on a regional to global scale. In this context, the Geostationary Environmental Monitoring Spectrometer (GEMS) onboard the Geostationary Korea Multi-Purpose Satellite-2B (GEO-KOMPSAT-2B), which was launched in 2020 by the Republic of Korea, aims for near real-time monitoring of air quality over Asia (Kim et al., 2020) where air quality is one of the biggest environmental health risks (Hopke et al., 2008). As the first ultraviolet (UV)–visible instrument in a geosynchronous orbit, GEMS can provide more detailed and frequent air quality data than existing low Earth orbit platforms. Since the first release of the GEMS data, some verification of its initial air pollutant products including nitrogen dioxide or ozone has recently been performed (e.g. Baek et al., 2023; Kim et al., 2023; Ghahremanloo et al., 2024). However, data validation and applications of many GEMS products are still largely lacking.

35

We focus on the GEMS aerosol optical depth (AOD), which measures the degree of light scattering or absorption at a given wavelength due to the presence of aerosols in the atmospheric column (Chudnovsky et al., 2012). Satellite-derived AOD serves as a crucial proxy for understanding aerosol distribution and its impact on air quality. However, the accuracy of satellite AOD data needs to be validated to ensure their reliability for downstream applications, including ground-level PM estimation. Typically, this involves comparisons with ground-based measurements (e.g. Ogunjobi and Awoleye, 2019; Mangla et al., 2020). For instance, Choi et al. (2019) evaluated various satellite-derived AODs against ground-based AOD measurements collected during the 2016 KORUS-AQ campaign in East Asia. Similarly, Cho et al. (2024) validated the performance of GEMS aerosol products against ground measured data. Both studies revealed a good correlation between satellite and ground-based AOD data, demonstrating the utility of satellite-derived AOD for monitoring data-scarce regions.

45

Here we first evaluate GEMS AOD data through a direct comparison with ground-based Aerosol Robotic Network (AERONET) observations over South Korea. However, we place greater emphasis on evaluating the utility of GEMS AOD for estimating ground-level PM concentrations, as it offers a unique opportunity to address aerosol data gaps in Asia (Wen et al., 2023). Moreover, South Korea has nationwide air quality monitoring stations, allowing us to obtain continuous and large data samples (PM10 and PM2.5) for validating the satellite data. To better utilise GEMS AOD for ground-level PM estimation, we employ machine learning models, which offer the advantage of experimenting with a wide range of input variables. For example, ground-level aerosols are not related to AOD only, but influenced by meteorological conditions or precursor pollutants such as sulfur dioxide ($SO_2$) and nitrogen dioxide ($NO_2$). Machine learning allows for the efficient integration and processing of these diverse datasets, enhancing the ability to utilize AOD for aerosol estimation.

55

Satellite-derived AOD has already been widely used to predict ground-level PM concentrations (Shin et al., 2020), as can be seen in the example of Moderate Resolution Imaging Spectroradiometer or Geostationary Operational Environmental Satellite (Gupta et al., 2006; Chudnovsky et al., 2012; Yang et al., 2020b; Zhai et al., 2021; Hammer et al., 2023). Nonetheless, inconsistent relationships between satellite-derived AOD and ground-level PM observations have been reported among different regions and based on data from different satellite instruments (Yang et al., 2020b). Therefore, there is an urgent need to evaluate the effectiveness of GEMS AOD data in estimating PM concentrations over Asia, which can in turn provide initial guidance for both data users and algorithm developers.

Consequently, our study aims to demonstrate the usefulness of GEMS AOD in PM modelling and highlight limitations in the current version of the data. To achieve this, we use GEMS AOD data over South Korea during the first two years of observations from January 2022 through to December 2023 (the very first data were available from November 2021). To estimate surface PM concentrations using AOD, additional input variables (meteorological data) are obtained from reanalysis data. For the model, we employ the random forest (RF) and Extreme Gradient Boosting (XGBoost), which are widely used machine learning methods for PM estimation given their great flexibility and strong predictive performance (Ma et al., 2020; Shin et al., 2020; Hu et al., 2017; Guo et al., 2021). This approach considers the general application of GEMS AOD in other Asian regions within the GEMS observation coverage beyond South Korea. Reanalysis data are particularly advantageous for obtaining meteorological variables in regions lacking ground-based weather observations. Furthermore, machine learning models provide practical benefits, as they can be applied to different regions without requiring extensive model parameter adjustments compared to physical models. In addition, we assess the potential improvement in machine learning model performance by incorporating supplementary surface chemical data available in South Korea. The results of this analysis are reported in detail.

In the following sections, we first describe the data and its preprocessing in Sect. 2. Section 3 details the methodology, including the machine learning models employed for PM estimation. In Sect. 4, we present the results and discuss their implications, followed by conclusions and future research directions in Sect. 5.

## 2 Data

The hourly PM concentration data in South Korea for the period of January 2022 to December 2023 used in this study are obtained from the real-time ambient air quality monitoring system, called AirKorea (https://www.airkorea.or.kr/, operated by the Korea Environment Corporation, a government-affiliated public institution under the Ministry of Environment. The PM concentrations are determined using a $\beta$-ray absorption method (Hauck et al., 2004), and the measurements have undergone quality controls to remove anomalous values before the release of the final data. Out of more than 600 ground-based AirKorea stations, we select a total of 499 urban air quality monitoring stations to represent human exposure to PM. While the stations are distributed across the country, a large number of stations are concentrated in the densely populated Seoul Capital Area.

The GEMS on board the GEO-COMPSAT-2B satellite has been in operation since 2020. The GEMS instrument measures the UV-visible radiance spectrum, and its geostationary orbit enables AOD retrievals at an hourly frequency during cloud-free daytime conditions (Kim et al., 2020). The GEMS measures radiance in the 300–500 nm range with a spectral resolution of 0.6 nm and retrieves aerosol properties. The GEMS aerosol retrieval algorithm (AERAOD) uses the optimal estimation (OE) method, which integrates satellite-observed radiances with initial estimates of aerosol properties, including AOD, derived from the two-channel inversion approach employed by the OMAERUV algorithm (Torres et al., 2007). The GEMS aerosol products provide final AOD data at three wavelength channels with a nominal spatial resolution of 3.5 $km$ x 8 $km$ at Seoul. Further details about the GEMS aerosol retrievals can be found from NIER (2020) and Go et al. (2020). We use GEMS AOD Level 2 (L2) data (at 443 $nm$), extracted within a $\pm$ 15 min time window of the PM measurement times, and taken from the pixel nearest to the AirKorea monitoring stations, with an average distance of 2.03 $km$. However, GEMS data values are often missing at the closest pixel due to issues such as cloud contamination or sun glint, resulting in an average of 1,990 AOD-PM data pairs per station (see Fig. S1 in Supplementary). Note that GEMS provides hourly observations of AOD during the daytime, corresponding to six to ten times per day depending on the season.

The relationship between AOD and PM concentrations can be affected by meteorological conditions (Koelemeijer et al., 2006; Tian and Chen, 2010; Handschuh et al., 2022). We consider boundary layer height (BLH), relative humidity (RH), air temperature (TEMP), surface pressure (SP), and wind speed and direction (WS and WD, respectively) as input features, in addition to AOD, to estimate the PM concentrations using machine learning models. Those input variables are selected based on previous studies (Yang et al., 2020a; Handschuh et al., 2022), including Seo et al. (2015), which examines the importance of incorporating meteorological data for accurate PM estimation in South Korea using satellite-derived AOD. BLH data from the fifth-generation ECMWF reanalysis (ERA5) at a 0.25-degree resolution are employed as a proxy for the vertical aerosol concentration in the lower troposphere as AOD is assumed to represent attenuation in the boundary layer (Gupta and Christopher, 2009b). The other variables are all obtained from ERA5-Land at a 0.1-degree resolution. RH is computed using temperature and dew temperature (Lawrence, 2005), while wind speed and direction are calculated using u and v wind components. Both ERA5 and ERA5-Land data are available at an hourly temporal resolution (Hersbach et al., 2020). Both gridded datasets are interpolated to the locations of AirKorea stations using inverse distance weighting based on the four closest grid points. If data are missing in the nearest grid points (e.g. over ocean areas), the corresponding locations are excluded from the analysis.

We obtain input data from reanalysis datasets, which are readily available across all areas within the GEMS satellite observation coverage. This ensures that the experiment conducted in this study can be easily extended to other locations, including other Asian countries, particularly in areas where meteorological measurements are unavailable. Additionally, reanalysis datasets provide consistent and reliable data updates over space and time. Nonetheless, it is well known that gases such as carbon monoxide (CO), $NO_2$, and $SO_2$ can influence PM formation mechanisms either directly or indirectly (Lee et al., 2024a). Therefore, we also incorporate chemical data measured at the AirKorea stations as additional input features. In this way, we can evaluate the potential improvements in PM estimation using AOD when supplemented with additional information, and

we report the corresponding results. The input variables used in this study are listed in Table 1.

Finally, for direct comparison, we obtain ground-based AOD measurements from AERONET sites in South Korea. A total of nine stations are selected, where data are available during the study period. AERONET provides highly accurate AOD measurements using Cimel Electronique Sun–sky radiometers, with an uncertainty of approximately 0.01-0.02 (Eck et al., 2019; Giles et al., 2019). For this study, we use the version 3, level 2.0 quality-assured AOD at 440 $nm$. For the comparison, GEMS

AOD data within a 5 km radius of the AERONET sites are considered, and sub-hourly AERONET data are averaged within a temporal window of $\pm 20$ minutes around the GEMS observation time.

## 3  Methods

For the AOD-PM simulations, we employ an RF algorithm with 100 trees and train this algorithm using AOD and meteoro-

logical data as input features and ground PM measurements as the target variable, respectively, at each station. RF operates by constructing multiple decision trees during training and aggregating their predictions to enhance accuracy and avoid overfitting. It is widely recognized for its ability to efficiently handle non-linear relationships in data and is often used for estimating PM concentrations (e.g. Hu et al., 2017; Guo et al., 2021). We also use XGBoost, which is similarly based on decision trees. However, XGBoost builds trees sequentially, allowing each tree to learn from the errors of the previous one, and is generally

considered to outperform RF (Chen and Guestrin, 2016). To ensure computational efficiency, we optimize the hyperparameters of both models by testing the models on a randomly selected subset of 50 stations, representing 10% of the total stations.

The main analysis is based on model predictions obtained through five-fold cross-validation. Specifically, we randomly split the entire data into five subgroups and use four of them (80%) as training data and the remainder (20%) as validation data.

This process is repeated five times such that every data subset is used as validation data at least once. Consequently, while this approach provides robust estimates of model performance, the actual performance of a model trained on the entire dataset could be underestimated due to the reduced size of the training data during cross-validation.

RF or XGBoost can be relatively easy to implement compared to process-based models, which often require sophisticated

parameter calibrations. Moreover, they offer high flexibility by accommodating diverse data types and variables. This makes machine learning a useful tool for assessing the usefulness of new data, such as GEMS AOD. However, the primary disadvantage of machine learning is its 'black-box' nature, meaning we cannot fully understand why it produces certain estimations. To address this limitation and examine the role of the input features, we further use SHapley Additive exPlanations (SHAP) and quantify the relative importance of the considered input features on the model's estimations. SHAP is an explainable machine

learning method based on Shapley values, which measure the marginal contribution of each predictor to the model's output or estimation, by evaluating how the model output changes when a feature is included or excluded across all possible feature

combinations (Molnar, 2019; Lundberg et al., 2020).

## 4 Results and discussion

### 4.1 Evaluation of GEMS AOD retrievals

First, we directly compare the GEMS AOD data with ground-based AOD measurements from the AERONET (Giles et al., 2019). As shown in Fig. 1a, the temporal variations of AODs at each station exhibit overall good correlations, with Persons's r ranging from 0.68 to 0.89. When the entire time series from all AERONET sites are compared, the correlation remains strong (r-value = 0.77), although GEMS tends to underestimate AOD compared to the ground-based measurements, as indicated by the linear regression slope (slp = 0.66) in Fig. 1b. Furthermore, Fig. 1c demonstrates that this underestimation is consistent across most AERONET AOD ranges, although overestimation can also occur at low AOD values. A study on the early version of GEMS L2 algorithm prior to the launch of GEMS also reported high correlation but slight underestimation of GEMS AOD relative to AERONET (Kim et al., 2020). Recent studies using GEMS L2 data in Asia regions have reported similar findings (e.g. Cho et al., 2024; Jang et al., 2024).

AOD retrievals from satellite observations can be influenced by several factors, including surface reflectance estimation or aerosol model assumptions. For instance, distinguishing surface reflectance and aerosol scattering or absorption effect can be challenging under low aerosol loading conditions (Rudke et al., 2023). Issues such as cloud contamination or heterogeneous surface conditions can also introduce uncertainties in satellite-derived AOD (Handschuh et al., 2022). Cho et al. (2024) specifically compared GEMS and AERONET AOD measurements for the period from 2021 to 2022 in Asia and pointed out that the absence of region-specific aerosol type information in the GEMS aerosol model, along with inaccuracies in cloud-masking processes, may adversely impact the accuracy of GEMS AOD data. Overall, the performance of GEMS AOD is comparable to ground-based measurements, confirming its potential for applications such as estimating surface aerosol properties and supporting air quality studies.

### 4.2 Performance of PM estimation derived using GEMS AOD and machine learning models

Figure 2a shows the average PM10 concentrations at the ground stations during the study period, calculated only when GEMS AOD observation data are available (i.e. collocated PM and AOD data pairs). The average of measured PM10 across the stations is 40.96 $\mu g\ m^{-3}$, ranging from 24.91 to 65.12 $\mu g\ m^{-3}$, which is higher than actual PM10 averages during the same period when nighttime data are also included (Fig. S2). Overall, relatively high PM10 concentrations are observed in western regions, which are related to strong inflow from the continent due to the prevailing mid-latitude westerlies in South Korea (Lee

et al., 2019).

Next, we estimate ground-level PM10 concentrations using RF models, which are trained individually at each station using AOD values and meteorological variables as input features (See Methods). RF can effectively handle non-linear relationships between input features and target variables, making it a useful tool for air quality modelling applications. We also confirm that RF outperforms empirical linear models in estimating PM10 (Fig. S3); all model performances are evaluated through the five-fold cross validation. Although XGBoost demonstrates comparable performance to RF, the main analysis focuses on RF results, as the differences between the two models are minimal, further supporting the robustness of the machine learning-based modeling approach. Overall, the RF models demonstrate satisfactory performance in estimating PM10 concentrations. The spatial distribution of PM10 concentrations observed in the measurements is effectively described by the model estimates (Fig. 2b), with an average estimated PM10 value of 41.28 $\mu g\ m^{-3}$, ranging from 24.88 to 65.52 $\mu g\ m^{-3}$, which closely matches the average measured value. The Pearson's correlation coefficients between the measured and estimated PM10 is 0.66, on average, across all the stations (Fig. 2c). The model performance is stable across the stations, as indicated by the correlation values between 0.59 and 0.73 (10th and 90th percentiles, respectively) at most stations. Similarly, RF models effectively estimate PM2.5 concentrations (Fig. 2d to f), achieving an average correlation value of 0.70, with correlation values ranging from 0.64 to 0.74 (10th to 90th percentiles).

Figure 3 compares the entire data combined from all stations and their estimation errors for PM10 (upper panels) and for PM2.5 (lower panels). GEMS AOD and PM measurements shows weak correlations, with r-value of 0.20 for PM10 (Fig. 3a) and 0.33 for PM2.5 (Fig. 3d), indicating that the relationship between columnar AOD and ground-level PM is not straightforward. In contrast, the estimated PM10 values, derived using both AOD and meteorological variables, show a closer agreement with ground measurements, as evidenced by a higher r-value of 0.67 (Fig. 3b). However, the regression slope deviates from the one-to-one line, revealing certain biases in the PM10 estimations. Note that the axes are transformed to logarithmic scales for better visualisation. For PM2.5, the regression line exhibits a similar level of deviation (Fig. 3e), although the RF models perform slightly better in terms of correlation with ground measurements, achieving a higher r-value of 0.72.

To investigate these biases further, relative error is defined as the difference between the estimated and measured PM divided by the measured values. The relative error is computed across predefined ranges based on deciles of the measured PM (Figs. 3c and 3f). For PM10, the relative error transitions from positive (overestimation) to negative (underestimation) at approximately $50\,\mu g\ m^{-3}$, which corresponds to the point where the regression line crosses from above to below the one-to-one line (Fig. 3b). Despite these biases, most estimated PM values have relative errors close to zero. For instance, the median relative errors in the ranges between the 40th and 90th percentiles are within $\pm 0.3$ for PM10. A similar pattern is observed for PM2.5, with consistent trends in relative error distributions across the analysed ranges. Consequently, larger biases occur at the minimum and maximum extremes of the measured PM ranges for both PM10 and PM2.5. This behaviour can likely be attributed to the underrepresentation of extreme values in the dataset, as these occur less frequently. Furthermore, machine learning models

prioritise overall accuracy, often leading to larger biases in those extremes. Measurement errors in extreme PM values may also contribute to the observed biases, as uncertainties tend to be greater in these ranges.

Furthermore, we use SHAP to quantify the relative importance of the considered input features on the model's estimations. As SHAP is computed for individual observations, we take the mean of absolute SHAP values for each input variable across all predictions to explain its global feature contributions (Fig. 4). For both PM10 and PM2.5 estimations, AOD has a great influence on model performance, demonstrating the effectiveness of satellite information for ground aerosol simulations. The direct relationships between PM and meteorological variables are not easily characterised (see Fig. S4), but SHAP analysis indicates that temperature (TEMP) is the most meteorological influential predictors for both PM10 and PM2.5 estimations. Similar results are observed for XGB (Fig. S5). For instance, temperature can promote PM particle production by enhancing the photochemical reactions in the atmosphere (Gupta and Christopher, 2009a). It can also act as an indicator of seasonal variations in PM concentrations; for instance, an increase in emissions from combustion processes during the winter time can result in high aerosol loading, while aerosols can easily be removed by wet deposition during a rainy season in summer (Kim and Kim, 2020).

Among other meteorological variables, relative humidity (RH) emerges as one of the most important predictors, particularly for PM10 estimations. RH influences wet deposition of PM and also characterises seasonal variations in PM concentrations. For instance, in Korea, PM10 tends to increase due to wildfire emission during dry seasons or from relatively coarse particles transported by Asian dust in the dry spring season. On the other hand, boundary layer height (BLH) has a greater importance for PM2.5 estimations. Given that aerosols are primarily confined to the planetary boundary layer, BLH is a good proxy with which estimate the height of the aerosol layer (Lee et al., 2024b) and can help relate columnar satellite data to surface aerosol values (Handschuh et al., 2022; Gupta and Christopher, 2009a). The stronger importance of BLH for PM2.5 estimation suggests its effectiveness in capturing the vertical distribution of finer aerosols. Previous studies have suggested a strong relationship between BLH and PM2.5 due to well-recognised positive feedback effects in their formation (e.g. Su et al., 2017; Wang et al., 2019-), which further underscores the importance of BLH. The relationship between AOD and PM is highly sensitive to variations in BLH conditions, as noted in previous studies (Zheng et al., 2017). For instance, higher BLH facilitates greater vertical dispersion of aerosols, thereby reducing surface PM concentrations for a given AOD. While our machine learning approach inherently captures such complex interactions, future work could explore the explicit sensitivity of BLH with the AOD-PM relationship to improve physical interpretability.

We further compare the temporal evolution of hourly PM10 measurements and estimates in each month. The measured PM10 in Fig. 5a displays clear seasonal patterns, with high concentrations recorded during the winter and spring months and low concentrations in the autumn. While PM10 values in South Korea often show strong diurnal or semidiurnal cycles (Kim and Kim, 2020), these diurnal variations are not clearly shown in these PM10 composites from multiple stations, which contain many gaps due to the missing values in the satellite data. Nonetheless, relatively higher concentrations are observed before noon, following the morning rush hour. This may be related to stable atmospheric conditions caused by relatively low temperature

during this time of day, which suppress convective circulation. The high concentrations during the cold season (winter to early spring) can be attribute to increased fossil fuel combustion for heating combined with the stagnant weather conditions during this season (Wang et al., 2015). Particularly in March, the highest PM10 is associated with the long-range transport of Asian dust originating in the deserts of Mongolia and China (Lee et al., 2019, 2024b; Kim et al., 2017). These seasonal variations are also well captured in the estimated PM10 concentrations (Fig. 5b), and the overall pattern is in good agreement with that of the ground measurements. The spatial correlation between the measured and estimated PM10 concentrations (Fig. 5a and Fig. 5b, respectively) is 0.96. Yet, we observe substantial differences in the magnitude, as shown in Fig. 5c, with underestimation of high values during the winter and spring and overestimation of low values in the autumn. These patterns align with the bias observed in the data comparison shown in Fig. 3. We also observe similar seasonal biases for PM2.5 (Fig. 5f), but with smaller magnitudes.

## 4.3 Enhancing PM estimation and applications at ungauged locations

In this section, we conduct two additional experiments. First, we incorporate supplementary air quality data to evaluate the potential for improving model performance using this additional information. Second, we assess the model's ability to estimate PM concentrations at locations without ground-based measurements by utilizing input data from neighboring stations. This allows us to examine the effectiveness of satellite-derived AOD and reanalysis meteorological data in scenarios where target data are unavailable.

Figure 6 presents the results of the main analysis repeated with additional measured data from AirKorea stations, including ozone ($O_3$), CO, $SO_2$, and $NO_2$. The inclusion of chemical pollutant data leads to overall improvements in model performance for both PM10 and PM2.5. Compared to the main analysis using only AOD and meteorological data (Fig. 3), the correlations between measured and estimated PM concentrations improve from 0.67 to 0.74 for PM10 and 0.72 to 0.85 for PM2.5. Furthermore, the relative error magnitude decreases across all ranges of measured PM concentrations. This overall improvement is anticipated, as chemical pollutants are known to influence PM concentrations by acting as precursors to secondary aerosols (Ngarambe et al., 2021; Kang et al., 2022). SHAP analysis highlights the significance of pollutant data, particularly CO and $NO_2$ (see Fig. S6), while the importance of GEMS AOD remains prominent. Although CO is not directly related to PM formation, it is positively correlated with PM concentrations, serving as an indicator of combustion sources and influencing atmospheric chemistry through photochemical reactions (Fu et al., 2020; Ngarambe et al., 2021; Dai et al., 2023). Meanwhile, $NO_2$ contributes to the formation of nitrate aerosols, and Lee et al. (2024a) shows that high PM2.5 concentrations in Seoul are strongly associated with elevated nitrate levels. While the use of globally available reanalysis meteorological data enhances the estimation of PM concentrations using GEMS AOD, these results demonstrate that incorporating additional locally available data can further improve model performance.

Finally, we examine the potential of satellite-derived AOD and machine learning models for estimating PM concentrations at ungauged locations. While South Korea has a relatively dense air quality monitoring network, these stations are primarily

concentrated in urban areas. And, many countries in Asia lack sufficient ground stations for comprehensive aerosol mapping. To simulate ungauged conditions, we retrain the RF model at each station by utilising data from its $n$ neighbouring stations, excluding any data from the target station itself. The model is then validated at the target station using a five-fold cross-validation approach, consistent with the main analysis.

Fig. 7 illustrates the PM estimation performance at ungauged locations, comparing correlation coefficients (Fig. 7a) and mean relative error (Fig. 7b). The 'gauged' case represents the model trained and tested using only the target station's data (i.e. the main results presented in the previous section). As expected, when the model is trained with data from a single neighboring station ($n = 1$), performance is lower than that of the gauged model due to limited regional representativeness. However, the models show higher correlation coefficients and decreased biases as more neighboring station data are included. This finding aligns with the understanding that machine learning models can effectively leverage spatial and temporal variability to enhance robustness and generalisability (O et al., 2020). When data from at least four neighboring stations are utilized, the model achieves PM estimation performance at ungauged locations comparable to that at gauged sites. However, it is important to note that in this study, neighboring stations are selected based solely on proximity without considering the similarity of conditions between stations. Future research should investigate how inter-station similarity impacts model performance, which may further optimize regionalisation approaches for aerosol and air quality modeling. In addition, incorporating more neighboring stations provides a larger amount of training data, meaning that the observed model improvement is driven not only by increased data diversity but also by the larger data size.

## 5 Conclusions

Applying satellite-derived AOD observational data to estimate ground-level PM offers an excellent opportunity for air quality monitoring, including at ungauged sites (Hammer et al., 2023; Filonchyk et al., 2020; Wei et al., 2023). This is particularly important for Asian regions, where a significant proportion of the population is exposed to air pollution levels exceeding WHO guideline values (Cohen et al., 2017). As the world's first geostationary earth orbit environmental instrument, GEMS is expected to provide more detailed air quality information over Asia with higher spatial and temporal resolutions than existing low Earth orbit platforms. The GEMS will also join a constellation of geostationary air quality satellites, together with TEMPO over North America and Sentinel-4 over Europe, to collectively provide near-global coverage (Kim et al., 2020). In line with ongoing efforts to confirm the reliability of the new satellite data products, in this study, we evaluate the effectiveness of GEMS AOD data by modelling AOD-PM relationships at around 500 stations in South Korea using machine learning models.

In this study, we aim to assess the potential of GEMS AOD data for sub-daily PM estimation using practical ML models, without prioritising the achievement of optimal model performance. While more advanced ML techniques or alternative modeling approaches (e.g. chemical transport models) could enhance performance, they are beyond the scope of this work. Furthermore, to our knowledge, this is the first evaluation of GEMS AOD applications, providing baseline results for future

model comparisons and development. This approach also applies to input selection. While we consider a relatively wide range of input variables, including both meteorological and chemical data, additional variables can be tested, and model performance can be compared. In this context, ML models are particularly advantageous, as they can incorporate a broad spectrum of variables, including those typically not used in process-based models. However, ML performance is highly dependent on the quality of the training data; therefore, careful attention to data quality is essential. Here we use reanalysis data as input features in addition to the satellite data to ensure the general applicability of ML models over diverse locations. Since reanalysis data are gridded, there is a spatial mismatch between the grid pixel and the actual ground target points (i.e. the AirKorea stations). Although this distance is within a few kilometers after applying the interpolation method to the reanalysis data, and no significant performance degradation due to this discrepancy is observed (Fig. S7), employing higher-resolution data or improved interpolation techniques for data processing could also be considered in the future.

Our results demonstrate that GEMS AOD, combined with meteorological variables such as temperature and boundary layer height, enables the model-estimated PM concentrations to achieve strong temporal and spatial correlations with ground measurements. However, biases between the estimated and measured PM concentrations are evident, particularly at extreme ranges, which may be attributed to limitations in both the models and the data. This limitation should be carefully addressed in future satellite retrieval algorithms and data applications. We also show that the model's performance could be improved by acquiring additional training data. The AirKorea stations provide not only PM measurements but also data on air-quality-related pollutants. Utilizing these additional measurements reduces ML estimation errors over most stations. Beyond the pollutant measurements in target locations, other PM-related data, such as information on pollution sources (e.g. emission levels or ambient air conditions at emission sites), could also enhance model performance. Future studies should assess the feasibility and contribution of acquiring such additional data on a region-specific basis. Machine learning is also known to regionalise effectively using training data obtained from regions with environmental conditions similar to the target region. Our study demonstrates that PM estimations at ungauged sites are possible by leveraging training data from nearby stations. In these cases, ungauged sites refer to regions lacking ground observations. Future efforts could explore the use of satellite data (AOD) from these ungauged areas, for instance, by fine-tuning ML models, enabling more precise and accurate predictions.

Ultimately, our study demonstrates the applicability of the current version of GEMS AOD data for estimating ground-level PM and provides valuable baseline information for ongoing improvements in data and modeling techniques. This research highlights the significant potential of GEMS AOD data to enhance air quality monitoring across Asia, particularly in regions where conventional ground-based measurements are unavailable or limited. By establishing a foundation for future studies and offering insights into model improvement strategies, our work underscores the importance of employing geostationary satellite data in air quality assessments.

*Code availability.* Code supporting this paper is published online at https://github.com/osungmin/gems_aod.

*Data availability.* GEMS AOD data can be requested from the Environmental Satellite Center website (https://nesc.nier.go.kr/). PM measurement data is publicly available from the AirKorea website (https://www.airkorea.or.kr/). ERA5 and ERA5-Land can be freely downloaded

from the Climate Data Store of the Copernicus Climate Change Service (https://cds.climate.copernicus.eu/).

*Author contributions.* SO designed the study, performed the experiments, and drafted the manuscript. JWY and SKP discussed the results and contributed to the writing.

*Competing interests.* The authors declare that they have no conflict of interest.

*Acknowledgements.* This research is supported by the Specialized University Program for Confluence Analysis of Weather and Climate

Data of the Korea Meteorological Institute (KMI) funded by the Korean government (KMA), with participation from Kangwon National University and Ewha Womans University. SKP acknowledges the National Research Foundation of Korea (NRF) grant funded by the Korea government (MSIT) (2021R1A2C1095535) and partly by the Basic Science Research Program through the NRF funded by the Ministry of Education (2018R1A6A1A08025520). The authors thank the principal investigators of each AERONET site for their efforts in establishing and maintaining the AERONET sites (https://aeronet.gsfc.nasa.gov/) the data of which are used in the study.

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

**Table 1.** List of input and target data used in PM modeling

| Category | Name | Description | Data Source |
|---|---|---|---|
| **Input Data** | | | |
| Aerosol Data | AOD | Aerosol Optical Depth | GEMS |
| Meteorological | BLH | Boundary Layer Height | ERA5 |
| | RH | Relative Humidity | ERA5-Land |
| | TEMP | Air Temperature | ERA5-Land |
| | SP | Surface Pressure | ERA5-Land |
| | WS | Wind Speed | ERA5-Land |
| | WD | Wind Direction | ERA5-Land |
| Chemical | CO | Carbon Monoxide | AirKorea |
| | $SO_2$ | Sulfur Dioxide | AirKorea |
| | $NO_2$ | Nitrogen Dioxide | AirKorea |
| | $O_3$ | Ozone | AirKorea |
| **Target Data** | | | |
| Particulate Matter | PM10 | Particles with a diameter of 10 micrometers or less | AirKorea |
| | PM2.5 | Particles with a diameter of 2.5 micrometers or less | AirKorea |

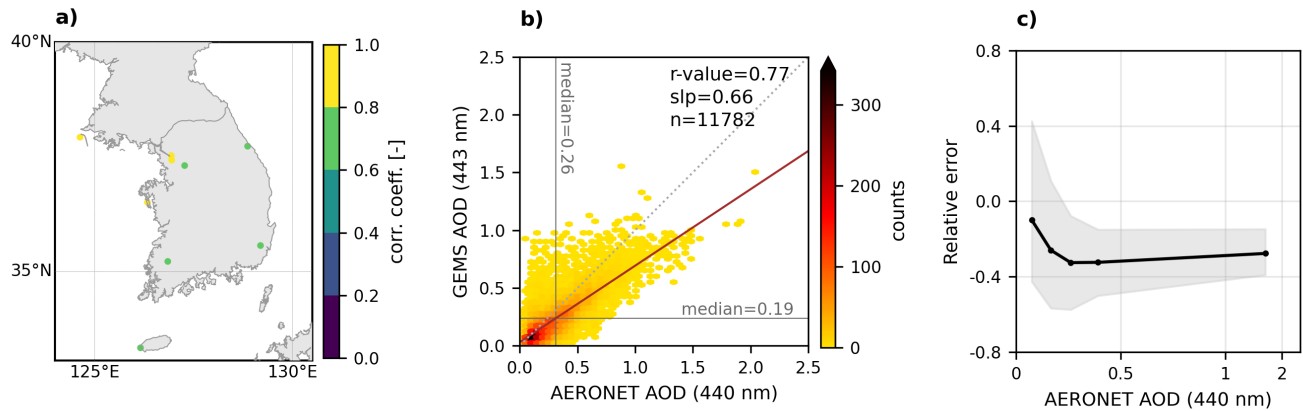

**Figure 1. Comparison between GEMS AOD and AERONET AOD observations**. (a) Correlations between GEMS AOD (443 $nm$) and AERONET AOD (440 $nm$) at individual AERONET stations. (b) Density scatter plot between GEMS and AERONET AOD across all stations. The vertical and horizontal lines represent the corresponding median values. The thick solid line is the regression line, and the dotted diagonal line is the one-to-one. (c) Average relative errors, defined as the difference between GEMS AOD and AERONET AOD divided by AERONET AOD, are shown for different AERONET AOD ranges. These ranges are divided based on each 20th percentile of AERONET AOD. Shaded areas indicate the interquartile range of the errors within each range. The map is created using the Matplotlib basemap v1.2.2 toolkit.

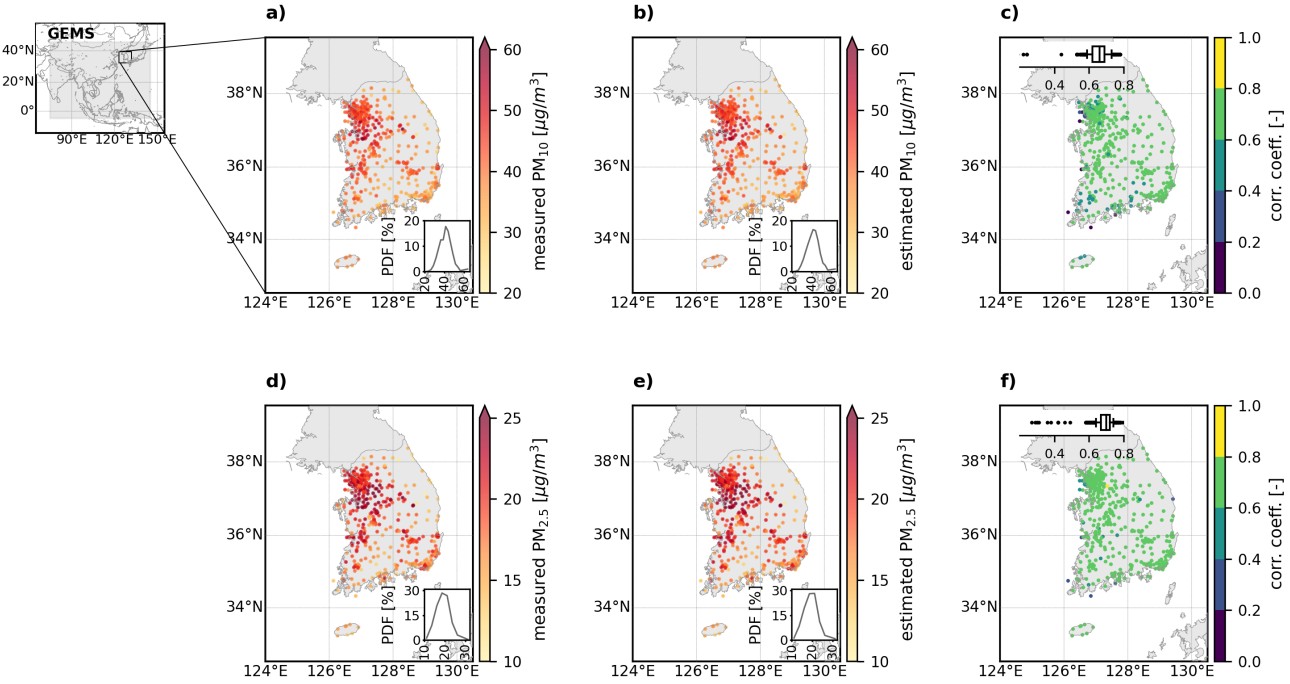

**Figure 2. Measured and model-estimated PM concentrations over South Korea during January 2022–December 2023.** The grey area in the small map shows the GEMS coverage (5°S–45°N, 75°E–145°E), with South Korea marked by a black box. Panels (a) and (b) present the average measured and estimated PM10 concentrations, respectively, and (c) shows their correlation. Insets in (a) and (b) display the probability density function (PDF), and (c) shows the correlation distribution via a boxplot, where the vertical center line is the median, and whiskers represent the 0.1 to 0.9 quantiles. Only data pairs with both PM and GEMS AOD values are included. Panels (d), (e), and (f) correspond to (a), (b), and (c), respectively, but for PM2.5.

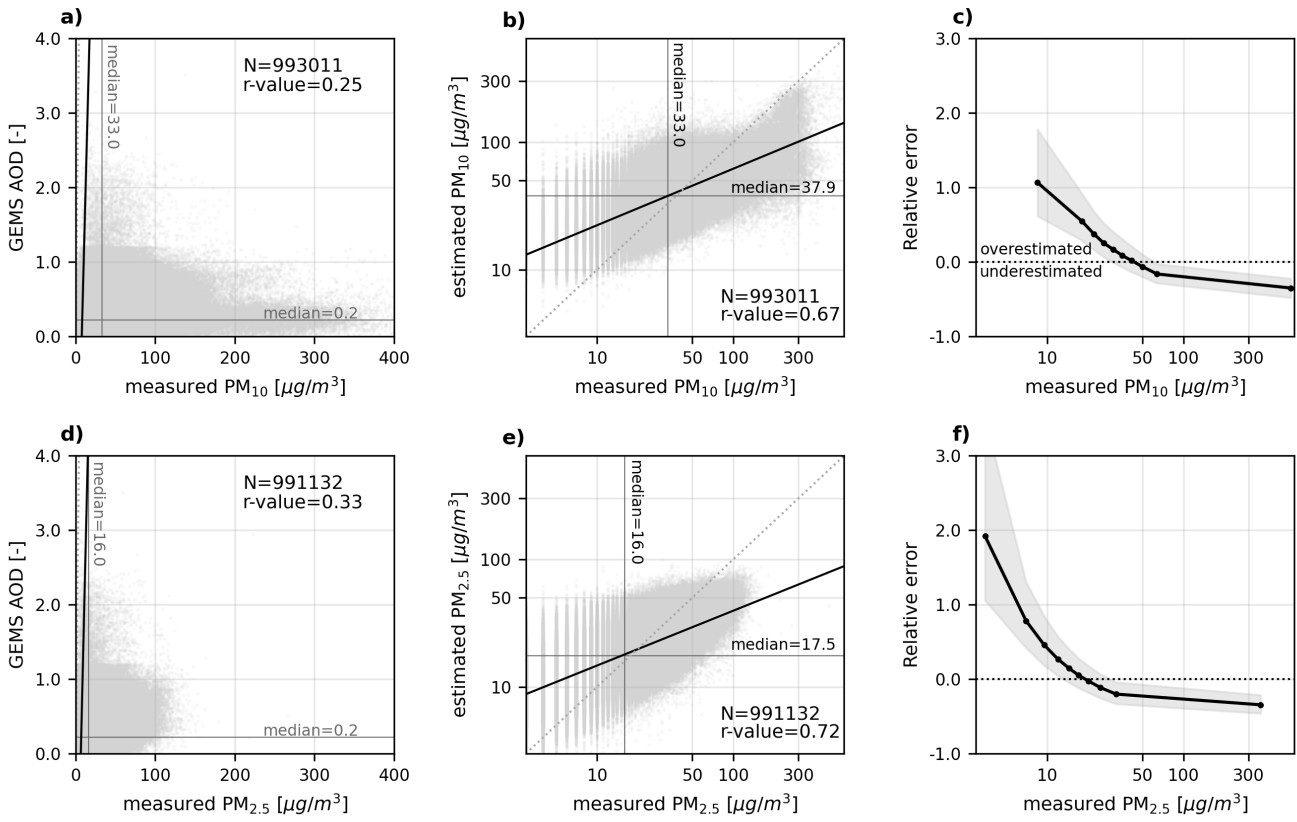

**Figure 3. Performance of RF models in estimating ground-level PM concentrations.** (a) Density scatter plot between measured PM10 and GEMS AOD across all stations. (b) Density scatter plot between measured PM10 and model-estimated PM10. The vertical and horizontal lines represent the corresponding median values. The thick solid line is the regression line, and the dotted diagonal line is the one-to-one. Both axes are displayed in log scale for better visualization. (c) Average relative errors, defined as the difference between estimated and measured PM divided by the measured values, are shown at predefined ranges using each decile of measured PM10. Shaded areas indicate the 25th-75th percentiles within each range. The x-axis is in log scale. Panels (d), (e), and (f) correspond to (a), (b), and (c), respectively, but for PM2.5.

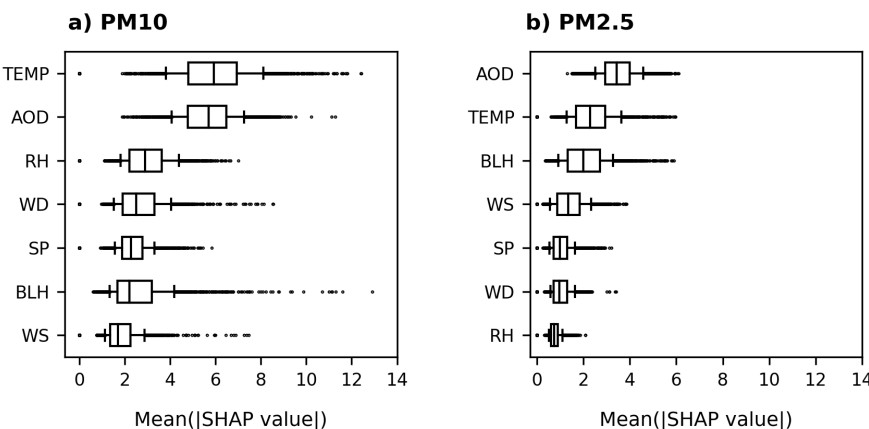

**Figure 4. Input importance of RF models.** SHAP values are computed to examine the contribution of each input feature to individual predictions for (a) PM10 and (b) PM2.5. In this box plot, the relative importance of the input variables is shown by ranking the averaged absolute SHAP values. The box represents the interquartile range, the vertical centre line is the median, and the whiskers represent the 0.1 to 0.9 quantiles, with outliers shown as individual dots.

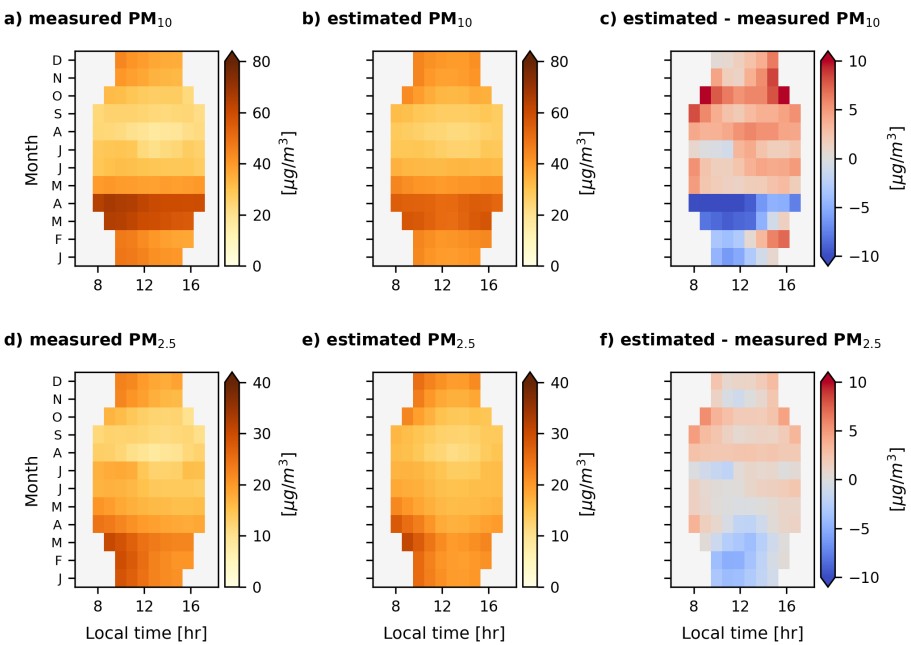

**Figure 5. Time-month diagram of PM measurements and estimates.** The mean hourly (a) measured and (b) estimated PM10 concentrations for each month averaged across all the stations. (c) represents differences between the PM10 measurements and estimates. Panels (d), (e), and (f) correspond to (a), (b), and (c), respectively, but for PM2.5.

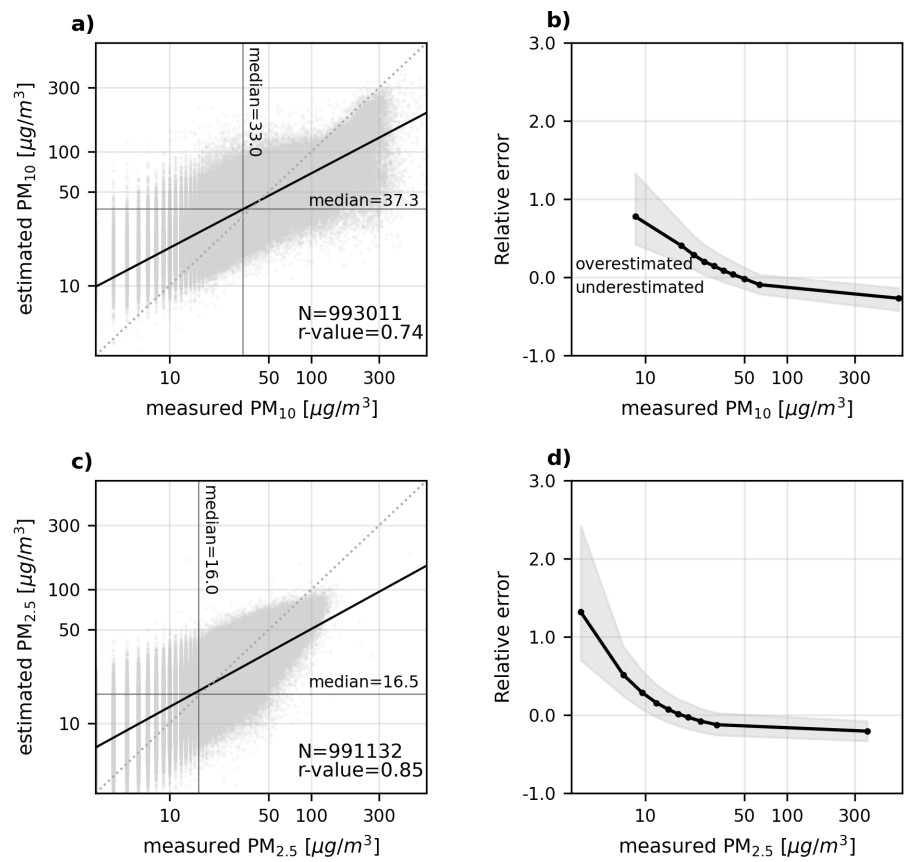

**Figure 6. Performance improvement of RF models with the inclusion of pollutant data.** This figure is similar to Fig. 3, but shows results for models trained with additional locally available data ($O_3$, CO, $SO_2$, $NO_2$) from AirKorea stations. (a) Density scatter plot between measured PM10 and model-estimated PM10. (b) Relative errors are shown at each 10th percentile of measured PM10. Panels (c) and (d) correspond to (a) and (b), respectively, but for PM2.5.

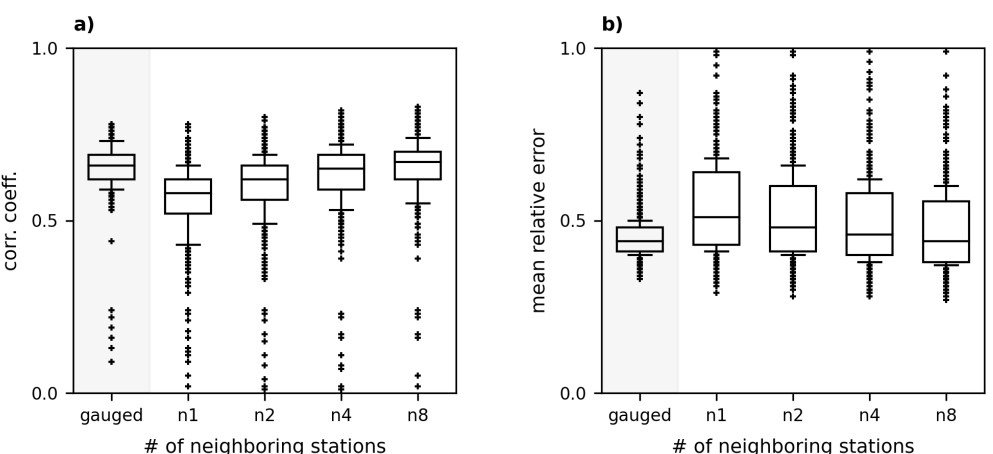

**Figure 7. Potential of satellite data in PM estimation at ungauged locations.** (a) Correlation and (b) mean relative error between the measured and estimated PM10 concentrations at each station. The RF models are trained using data from the $n$ closest neighboring sites, excluding the target station, where model performance is evaluated. The term 'gauged' indicates that the model is trained and tested at the same station (as shown in the main analysis in Fig. 2).