# Peer review of "Estimating hourly ground-level aerosols using GEMS aerosol optical depth: A machine learning approach"

_Atmospheric Measurement Techniques, 2024_

## Author Comment (AC1)

**Reply to Reviewer #1 for Manuscript of "Estimating hourly ground-level aerosols using GEMS aerosol optical depth: A machine learning approach" by O et al.**

The manuscript is based on the estimation of PM2.5 and PM10 from GEMS AOD. The main objective is to evaluate the effectiveness of GEMS AOD in estimating ground level PM concentrations. This study attempts to study how GEMS AOD can provide air quality estimates in a global scale, which is of great importance

⇨ *We thank the reviewer for recognizing the value of our study and for their thoughtful comments and efforts in reviewing our manuscript. Below, we provide point-by-point responses to the comments.*

⇨ *Here, we summarize the main revisions planned for the manuscript:*

- *Data Update: The dataset will be updated to include data through December 2023, resulting in two years of data being analyzed.*

- *Additional Machine Learning Algorithm: To enhance the robustness of our machine learning-based modeling, we will employ an additional algorithm, XGBoost (XGB).*

- *Expanded Input Features: We will include additional chemical gas features, such as $SO_2$, $NO_2$, and $O_3$, in the modeling process.*

- *Revised Manuscript Structure: The manuscript structure will be reorganized. Specifically:*
  - *The Data and Methodology sections will be separated.*
  - *The comparison results between GEMS AOD and AERONET AOD will be presented first, followed by the PM estimation results.*

However, there are few concerns regarding the formulation of the study and the structure of the manuscript. Given below are my suggestions.

- The overall paper lacks adequate explanations and citations to corroborate the objective of the study and how it differs from existing studies/novelty. (Ex: are there any ML based studies for estimation of PM concentrations? What are the advantages of this method over the existing?)

  ⇨ *To address the concern, we will enhance the introduction section to clearly state the objective of the study and highlight its novelty. We will also include a discussion on existing machine learning-based studies for PM concentration estimation and emphasize the advantages of our approach compared to previous methods.*

- The introduction of the manuscript should include a brief description on the sections of the manuscript. Results and discussion should be a separate section from data and methodology. I suggest separating data and methodology as separate sections, as this manuscript lacks proper description on the methodology (there is too little information on the machine learning method (RF), selection criteria for input variables, ranges of the input variables.

  ⇨ *Thank you for the suggestion. We will reorganize the sections as recommended and provide additional explanations to clarify the methodology.*

- What is the sample size of the data used in RF?

  ⇨ *Since RF models are applied to individual station points, the size of the training data varies for each model (station). To illustrate this, we will add a figure showing the distribution of training data sizes across all stations.*

- RF was selected to estimate PM concentrations out of some other ML methods. How do you evaluate the model effectiveness in this work? Model performance can also affect the conclusions you draw regarding the ability of GEMS AOD to accurately provide PM concentrations.

  ⇨ *We will provide a more detailed explanation of why RF was selected over other machine learning methods and elaborate on its effectiveness in this study. Additionally, we will employ another machine learning method, XGBoost, to compare the performance of different models and further validate our approach.*

- The first part of the results should be to validate the GEMS AOD retrievals

  ⇨ *We will reorganize the section to present the comparison between GEMS AOD and AERONET AOD first.*

- The labeling of PM measurements used in RF, and the PM estimations, is vague. Make it more distinct.

  ⇨ *Okay, we will revise the labeling to make the PM measurements and the PM estimations more distinct.*

- The use of mean vs error plots would be a better way of understanding the model performance rather than comparing the correlation coefficients. (Refer, Bland-Altman analysis)

  ⇨ *We will consider additional metrics to evaluate the model's performance from a broader perspective.*

- Add more details description on SHAP analysis.

  ⇨ *We will add more description on SHAP.*

- L 59-61 Include more details about GEMS instrument (uncertainties, wavelength channels). Do you perform any pixel averaging?

- ⇨ *We will add more information in the data section. We do not perform any pixel averaging to GEMS. This point will be further explained.*

- You need to add a description on GEMS AOD retrieval algorithm and explain possible uncertainties in AOD retrievals. The citation is not enough.

  - ⇨ *Thank you for pointing this out. We will add a detailed description of the GEMS AOD retrieval algorithm and discuss the possible uncertainties in GEMS AOD retrievals.*

- L 74 – Do you perform any geolocation of data? How do you collocate reanalysis data? What is the maximum possible difference in the colocation of ground-based PM concentrations, AOD and reanalysis data?

  - ⇨ *We will add more explanation on the geolocation of data (Inverse Distance Weighting).*

- L 115 – Is this something evident across all AOD values? Are there any differences seen for PM estimations under lower AOD and higher AOD values. Is there any detection limit?

  - ⇨ *To address the reviewer's concern, we will conduct additional analyses to illustrate how the model's performance varies under different conditions.*

- L 154 – AERONET data should be introduced under the data section. How do you collocate AERONET data? What do you mean by closest? You should specify the distance limit. How do you average temporal data. Does the difference between AERONET AOD and GEMS AOD lies with AEROENT AOD uncertainty?

  - ⇨ *We will include a detailed description of the AOD data and the preprocessing steps involved.*

- L 63 – What is ARA?

  - ⇨ *Aerosol Retrieval Algorithm, we will explain this.*

- L 69 – What is ERA5?

  - ⇨ *ECMWF Reanalysis v5 (ERA5) data, we will explain this.*

- L 75 – How did you perform the AOD-PM simulations? Or do you mean estimations?

  - ⇨ *We meant the PM estimation through the model simulations, we will make it clear.*

- L 83 – 89 This paragraph should go under the data section

  - ⇨ *Thank you for the suggestion. We will modify it.*

- L 157 – Has it been observed for low AOD or high AOD?

  ⇨ *We will add more details.*

- L 164 -165 Does the GEMS AOD algorithm consider any non-sphericity dust? You should add a description about the AOD algorithm

  ⇨ *We will add more description on the AOD algorithm*

- L 176 – Fig 6. What does n=1,2,… stand for?

  ⇨ *It refers to the number of stations providing the training dataset. We will clarify this explanation in the manuscript.*

---

## Author Comment (AC4)

**Reply to Reviewer #1 for Manuscript of "Estimating hourly ground-level aerosols using GEMS aerosol optical depth: A machine learning approach" by O et al.**

This manuscript is purposed on the PM concentration estimation based on the GEMS AOD observation data.

Because the GEMS is the geostationary orbit satellite, the idea in this manuscript has advantage of diurnal change monitoring of PM concentration from the satellite measurement.

⇨ *We thank the reviewer for recognizing the value of our study and for their thoughtful comments and efforts in reviewing our manuscript. Below, we provide point-by-point responses to the comments.*

⇨ *Here, we summarize the main revisions for the manuscript:*
- *Data Update: The dataset has been extended to December 2023, covering two years.*
- *Additional Machine Learning Algorithm: To enhance the robustness of our machine learning-based modeling, we have employed an additional algorithm, XGBoost (XGB).*
- *Expanded Input Features: We have considered additional chemical gas features, such as $SO_2$, $NO_2$, and $O_3$, in the modeling process.*
- *The manuscript structure is reorganized. Specifically:*
- *Data and Methodology sections are separated.*
- *Results with PM2.5 are reported in the main text.*

However, to understand and check this characteristics, this manuscript needs to some additional analysis. Especially, the machine learning method is not a perfect approach and its result can be changed by the input data selection.

For this reason, idealizing and analyzing the input variable for the machine learning method is essential to include the manuscript.

⇨ *We appreciate the reviewer's insightful comment. We agree that understanding the characteristics of the machine learning (ML) approach and its sensitivity to input data selection is crucial. To address this concern, we have conducted additional analyses focusing on the input data. Nonetheless, we would like to clarify that our primary aim is not to develop the best model for estimating PM concentrations but rather to evaluate the potential of new satellite data (GEMS AOD) as a valuable input for PM estimation. In this context, ML serves as an optimal tool due to its flexibility and ease in testing new data alongside other inputs that may contribute to improving model performance.*

For the detail, I listed to the below.

1) Introduction: For the readability of the manuscript, the author will be added to the brief explanation of sections in the final part of the Introduction section.

⇨ *Thank you for the suggestion. We have added the description on the sections in the end of the introduction:* *In the following sections, we first describe the data and its preprocessing in Section 2. Section 3 details the methodology, including the machine learning models employed for PM estimation. In Section 4, we present the results and discuss their implications, followed by conclusions and future research directions in Section 5.*

2) Adding the reference

- Because the study of satellite retrieved AOD was largely evaluated, the reference and related paragraph will be added before the paragraph of Line 33 (Before GEMS AOD study)

⇨ *We have added the following text in Introduction:* *[...] the accuracy of satellite AOD data needs to be validated to ensure their reliability for downstream applications, including PM estimation. Typically, this involves comparisons with ground-based measurements (e.g. Ogunjobi and Awoleye, 2019; Mangla et al., 2020). For instance, Choi et al. (2019) evaluated various satellite-derived AODs against ground-based measurements collected during the 2016 KORUS-AQ campaign in East Asia. Similarly, Cho et al. (2024) validated the performance of GEMS aerosol products against ground measured data. Both studies revealed a strong correlation between satellite and ground-based AOD measurements, demonstrating the utility of satellite-derived AOD for monitoring data-scarce regions.*

- L35: The references related to AOD definition and its retrieval method will be added, such as Go et al. (2020).

⇨ *Thank you for the suggestion. We have added more references including Torres et al., (2007), Go et al. (2020), and Cho et al. (2024):* *The GEMS aerosol retrieval algorithm (AERAOD) uses the optimal estimation (OE) method, which integrates satellite-observed radiances with initial estimates of aerosol properties, including AOD, derived from the two-channel inversion approach employed by the OMAERUV algorithm (Torres et al., 2007; Cho et al., 2024). The GEMS aerosol products provide final AOD at three wavelength channels with a nominal spatial resolution of 3.5 km x 8 km at Seoul. Further details about the GEMS aerosol retrievals can be found from NIER (2020) and Go et al. (2020).*

- L56: What is 'Korea Environment Corporation'? Is that 'Korean Environmental Institute'?

⇨ *K-eco is a government-affiliated public institution under the Ministry of Environment in South Korea, and it is responsible for managing AirKorea. We have modified the text as "[...] from the AirKorea real-time ambient air quality monitoring system operated by the Korea Environment Corporation, a government-affiliated public institution under the Ministry of Environment."*

- L57: For the PM concentration measurement, the author will be added the related references.

⇨ *The following reference is added: Hauck, H., Berner, A., Gomiscek, B., Stopper, S., Puxbaum, H., Kundi, M., and Preining, O.: On the equivalence of gravimetric PM data with TEOM and beta-attenuation measurements, J. Aerosol Sci., 35, 1135–1149, doi:10.1016/j.aerosci.2004.04.004, 2004.*

- L63: What is ARA? Need to clarify.

⇨ *Aerosol Retrieval Algorithm. We have clarified this in the reference list as "Algorithm Theoretical Basis Document (ATBD) for the GEMS aerosol retrieval algorithm."*

3) From the Section 2, this manuscript included both methodology and result parts. I suggest that this section separates the method section (e.g., Section 2) and Result section (e.g. Section 3), and the author will make sub-sections for the detailed explanation of each parts. In this version of the manuscript, method and result parts are too short to clarify the detailed machine learning method and the reason of variable selections. For this reason, the manuscript is not able to identify the difference of research compared to the several previous studies for PM estimation by the machine learning. The author have to include the table for the list of the selected variable and selection criteria.

⇨ *Following the reviewer's suggestion, we have restructured the manuscript to separate the methodology and results sections for greater clarity. The manuscript is now organized into Section 2: Data, Section 3: Methods, and Section 4: Results. Additionally, Section 4 has been divided into three subsections for 1. Direct comparison between GEMS and Aeronet AOD, 2. Use of GEMS AOD in estimating ground PM, and 3. Improving ML-based PM estimation.*

⇨ *To address the reviewer's comment, we have explained the reason for the variable and data selections in Sect. 2, with a table listing the selected variables: Those input variables are selected with reference to previous studies (Yang et al., 2020; Handschuh et al. 2022), including Seo et al., (2014), which conducted experiments in South Korea. [...] We obtain input data from reanalysis datasets, which are readily available across all areas within the GEMS satellite observation coverage. This ensures that the experiment conducted in this study can be easily extended to other locations, including other Asian countries, particularly in areas where meteorological measurements are unavailable. Additionally, reanalysis datasets provide consistent and reliable data updates over space and time. Nonetheless, it is well known that gases such as CO, $NO_2$, and $SO_2$ can influence PM formation mechanisms either directly or indirectly (Lee et al., 2024). Therefore, we also incorporate chemical data measured at the AirKorea stations as additional input features. In this way, we can evaluate the potential improvements in PM estimation using AOD when supplemented with additional information, and we report the corresponding results. The input variables used in this study are listed in Table 1.*

*Table 1. List of data used in PM modeling*

| Category | Name | Description | Data source |
|---|---|---|---|
| Input data | | | |

| Aerosol data | AOD | aerosol optical depth | GEMS satellite |
|---|---|---|---|
| | | | |
| Meteorological | BLH | boundary layer height | ERA5 reanalysis |
| | RH | relative humidity | ERA5-Land |
| | TEMP | air temperature | ERA5-Land |
| | SP | surface pressure | ERA5-Land |
| | WS | wind speed | ERA5-Land |
| | WD | wind direction | ERA5-Land |
| Chemical | CO | Carbon Monoxide | AirKorea station |
| | SO2 | Sulfur Dioxide | AirKorea station |
| | NO2 | Nitrogen Dioxide | AirKorea station |
| | O3 | Ozone | AirKorea station |
| Output data | | | |
| Particulate matter | PM10 | particles with a diameter of 10 micrometers or less | AirKorea station |
| | PM25 | particles with a diameter of 2.5 micrometers or less | AirKorea station |

4) L59-L64: For the data colocation between ground and satellite pixels, temporal colocation is clarfied in the manuscript. However, the spatial colocation between Airkorea and GEMS pixel, the manuscript is explained only the 'nearlest pixel'. How to be selected the 'nearlest pixel'? Because the cloud contamination of the GEMS AOD value, some pixels have to be eliminated, and the spatial distance between two measurements will be far. Do you have criteria of the maximum distance? In addition, the ground observation stations are dense in the urban region, especially denser than the GEMS spatial resolution. In this case, the same GEMS pixel is duplicately selected in different AirKorea observation sites. In this case, how to correct the colocation method?

⇨ *Please note that GEMS L2 data is provided on a pixel basis, and we selected the 'nearest pixel,' not the 'nearest observation.' This means that GEMS AOD values at the*

*nearest pixel are often missing due to issues like cloud contamination, leaving an average of 1990 data pairs (~11%) per station over the two-year period.*

⇨ *The average distance between GEMS pixels and AirKorea stations ranges from 1.5 to 2.7 km, while the average distance among AirKorea stations is 7.2 km. Therefore, we believe the impact of pixel duplication is minimal.*

⇨ *Further, we have confirmed that the spatial distances between GEMS pixels and ground stations do not significantly affect PM estimation performance, as shown in the figure below.*

[Figure]

***Figure S2.** PM estimation performance by distance between data sources and measurement stations. (a) and (b) show the distribution of distances between GEMS AOD observation pixels and PM measurement stations, along with average model performance (correlations) for PM10 and PM2.5, respectively. (c) and (d) depict the same, but for the distances between meteorological input data (i.e. ERA5-Land) and PM measurement stations.*

- In addition, for the colocation between observation and reanalysis dataset, what kind of interpolation method is used in this study? If you simply selected the 'nearlest grid', it may affect the uncertainty.

⇨ *We agree with the reviewer that simply selecting the "nearest grid" may introduce uncertainty. To address this, we have updated our colocation method to use Inverse distance weighting (IDW) is used. The updated method is explained as follows: Both gridded datasets are interpolated to the locations of AirKorea stations using inverse distance weighting (IDW) based on the four closest grid points. If data are missing in the nearest grid points (e.g., over ocean areas), the corresponding locations are excluded from the analysis.*

5) L132: Boundary Layer Height (BLH) is not a linear relation to correct the PM concentration from satellite AOD. The BLH is roughly changing the PM concentration. But its sensitivity is also changed by the columnar concentration of aerosols. Did the author check the sensitivity change of BLH for the relationship between PM concentration and satellite AOD? (Including the reference survey)

⇨ *In our machine learning approach, the model learns patterns and relationships directly from the data without explicitly assuming linear or non-linear relationships like the sensitivity of BLH. While the analysis of the sensitivity change of BLH is beyond the scope of our study, we acknowledge the importance of understanding these physical relationships.*

⇨ *To address this, we have expanded our explanation of the role of BLH as follows: "BLH is a good proxy with which to estimate the height of the aerosol layer and can help relate columnar satellite data to surface aerosol values. The relationship between AOD and PM is highly sensitive to variations in BLH conditions, as noted in previous studies (e.g. Zhang et al., 2016; Zheing et al. 2017). For instance, higher BLH facilitates greater vertical dispersion of aerosols, thereby reducing surface PM concentrations for a given AOD. While our machine learning approach inherently captures such complex interactions from the data, future work could explore the explicit sensitivity of BLH within the AOD-PM relationship to improve physical interpretability."*

6) Section 2.1: Although the supplement part include the PM2.5 result, the body of the manuscript is not shown the detailed analysis of PM2.5. I suggested that both the PM10 and PM2.5 estimation method and SHAP analysis will be included separately. Also, the detailed SHAP analysis results have to be included with the detailed analysis and explanations. If the author compares the difference between PM2.5 and PM10 estimation, it is possible to evaluate the contribution of aerosol types or absorptivity. In addition, for the explanation of PM10 concentration, the manuscript is confused about what is 'observed from AirKorea' and 'satellite retrieved PM concentrations'. The author will clarify the word for satellite-derived PM concentration and Ground-based observed PM concentration.

⇨ *We appreciate the reviewer's detailed feedback and suggestions. While it would be interesting to investigate aerosol types or absorptivity, SHAP values primarily indicate the importance of input features and do not provide direct evidence to analyze such detailed characteristics. Thus, we consider this aspect suitable for future research.*

⇨ *Following the reviewer's suggestion, we have included the results for PM25 in the main text, along with its SHAP analysis. Please note that a computational error in the SHAP analysis was identified and corrected, resulting in slight changes to the results. However, AOD remains one of the most important input predictors.*

⇨ *We have also expanded the explanation and discussion of the SHAP analysis as follows: "For both PM10 and PM25 estimations, AOD has a greater influence on model performance than most meteorological variables, as expected from its relatively strong correlation with ground aerosols (Fig. 2a), demonstrating the effectiveness of satellite information for ground aerosol simulations. The relationships between PM and meteorological variables are not straightforward (see Fig. S2), but SHAP analysis indicates that temperature (TEMP) is the most influential meteorological variable for both PM10 and PM25 estimations. [...] Similar results are observed for XGB (Fig S3).*

*Among other meteorological variables, relative humidity (RH) emerges as one of the most important predictors for PM10 estimations. RH influences wet deposition of PM and also characterises seasonal variations in PM concentrations. For instance, in Korea, PM10 tends to increase due to wildfire emissions during dry seasons or from relatively coarse particles transported by Asian dust in the spring season. On the other hand,*

*BLH has a greater importance for PM25. As aerosols are primarily confined to the planetary boundary layer, BLH is a good proxy for estimating the height of the aerosol layer (Lee et al., 2024) and can help relate columnar satellite data to surface aerosol values (Handschuh et al., 2022; Gupta and Christopher, 2009a). The stronger importance of BLH for PM25 suggests its effectiveness in capturing the vertical distribution of finer aerosols. Previous studies have suggested a strong relationship between BLH and PM25 due to well-recognized positive feedbacks (e.g. Li et al., 2017; Su et al., 2022), which further underscores the importance of BLH. The significant influence of both BLH and RH on the relationship between AOD and PM has also been reported by Zheng et al., 2017.*

⇨ *To address the confusion regarding terminology, we have carefully reviewed and revised the manuscript to clearly distinguish between satellite-derived and ground-based observed PM concentrations.*

7) Figure 4 and 5: Re-arrange the time scale (24 hours -> Daytime)

⇨ *We have revised Figure 4 to reflect daytime data only, as suggested. Please note that Figure 5 has been removed in the updated manuscript.*

8) L161-L170: The author mentioned that the main reason for estimated PM underestimation is due to the GEMS AOD underestimation. However, this study's method made the machine learning model based on the GEMS AOD. If so, the uncertainty characteristics of GEMS AOD is adopted in the machine learning modeling. Another possibility of the estimated PM underestimation is the false selection of variables or lack of the variable for the machine learning method. From several previous studies, the PM concentration is not affected only by the meteorological components, but also by the chemical processes. The author has to check the variable selections.

⇨ *We agree with the reviewer that machine learning models should account for biases inherent in GEMS AOD. Following the reviewer's suggestion, we conducted additional experiments by incorporating chemical data as input variables for PM simulations. These experiments demonstrated that including chemical data can indeed improve the model's performance. However, the primary aim of our study is to use 'commonly-available' input data, such as reanalysis meteorological data and GEMS AOD, to ensure the generalizability of our approach to other Asian countries, where ground measurements may not be readily available.*

⇨ *Therefore, we have reported the results of these additional experiments as a test analysis under Subsection 3.3. This highlights the potential for improved model performance when supplementary data, such as chemical pollutants from ground stations, are incorporated. In fact, while model performance improves with the inclusion of additional input data, discrepancies between the measured and estimated PM concentrations persist. These discrepancies could be attributed to limitations in the quality of AOD data (e.g., biases compared to ground-based AOD measurements) and/or inherent limitations of the machine learning models.*

⇨ *The new result is reported in Fig. 6.*

[Figure]

*Figure 6. Performance improvement of RF models with the inclusion of pollutant data. This figure is similar to Fig. 3, but shows results for models trained with additional locally available data ($O_3$, CO, $SO_2$, $NO_2$) from AirKorea stations. (a) Density scatter plot between measured PM10 and model-estimated PM10. (b) Relative errors are shown at each 10th percentile of measured PM10. Panels (c) and (d) correspond to (a) and (b), respectively, but for PM25.*

9) For the Machine learning adaptation, do you have the criteria of minimum concentration of observed PM and minimum value of satellite retrieved AOD? Low concentration of aerosol cases may be affecting the overall performance of estimation.

⇨ *Inspired by the reviewer's comment, we conducted additional experiments using ML models trained exclusively on high PM concentration cases. Unfortunately, this approach did not result in any noticeable improvement in model performance (see the figure below). Instead, we found that model performance can be enhanced by incorporating more informative input data, such as chemical pollutant measurements (see our responses to the comment above).*

⇨ *While there may be other approaches to further improve ML model performance, we consider this to be outside the scope of the current study. Our primary objective is to evaluate the usefulness of GEMS AOD in PM estimation and provide a baseline model performance for future studies.*

[Figure]

10) L179-L184 and Figure 6: For the statistical score, a detailed explanation will be needed. In addition, in Figure 6, 'n0', 'n1',' n2', 'n4', and 'n8' are not explained in the caption of Figure 6 and the body of the manuscript. The author has to clarify the explanation of Figure 6 and the mean of the statistical score.

⇨ *We apologize for the lack of clarity in the explanation. It refers to the number of neighboring stations providing the training dataset. We have updated the figure caption to clarify this. Additionally, please note that we have slightly modified the experimental setup to demonstrate the potential of satellite-derived AOD and machine learning for estimating PM concentrations at ungauged locations.*

[Figure]

*Figure 7. Potential of satellite data in PM estimation at ungauged locations. (a) Correlation and (b) slope of the linear regression between the measured and estimated PM10 concentrations at each station. Data from the n closest neighboring sites are used to train the RF models, and the model performance is evaluated at the target station, where the training data is deliberately excluded for this experiment. The term 'gauged' indicates that the model is trained and tested at the same station (as shown in the main analysis in Fig. 2).*

---

## Author Comment (AC5)

**Reply to Reviewer #2 for Manuscript of "Estimating hourly ground-level aerosols using GEMS aerosol optical depth: A machine learning approach" by O et al.**

The manuscript is based on the estimation of PM2.5 and PM10 from GEMS AOD. The main objective is to evaluate the effectiveness of GEMS AOD in estimating ground level PM concentrations. This study attempts to study how GEMS AOD can provide air quality estimates in a global scale, which is of great importance

⇨ *We thank the reviewer for recognizing the value of our study and for their thoughtful comments and efforts in reviewing our manuscript. Below, we provide point-by-point responses to the comments.*

⇨ *Here, we summarize the main revisions for the manuscript:*
  - ***Data Update:*** *The dataset has been extended to December 2023, covering two years.*
  - ***Additional Machine Learning Algorithm:*** *To enhance the robustness of our machine learning-based modeling, we have employed an additional algorithm, XGBoost (XGB).*
  - ***Expanded Input Features:*** *We have considered additional chemical gas features, such as $SO_2$, $NO_2$, and $O_3$, in the modeling process.*
  - ***The manuscript structure*** *is reorganized. Specifically:*
    - *Data and Methodology sections are separated.*
    - *Results with PM2.5 are reported in the main text.*

However, there are few concerns regarding the formulation of the study and the structure of the manuscript. Given below are my suggestions.

1. The overall paper lacks adequate explanations and citations to corroborate the objective of the study and how it differs from existing studies/novelty. (Ex: are there any ML based studies for estimation of PM concentrations? What are the advantages of this method over the existing?)

⇨ *We appreciate the reviewer's comment and have clarified the study's objectives and novelty. Our research is one of the first to validate GEMS AOD data, focusing on its utility for PM estimation rather than direct comparisons with ground-based AOD. Please note that we do not aim to develop best models for specific regions; instead, we evaluate GEMS AOD's broader applicability, such as improving models with additional data (e.g., chemical data) or estimating PM in areas without ground observations (e.g., using neighboring stations). In this context, we use ML models that are flexible enough to integrate diverse input data and can be relatively easily implemented in other regions, unlike physics-based models that often require additional parameter calibrations.*

⇨ *These points have been added to the introduction and conclusions for better clarity. e.g.* *Here, we first evaluate GEMS AOD data through a direct comparison with ground-based AERONET observations over South Korea. However, we place greater emphasis on evaluating the utility of GEMS AOD for estimating ground-level PM concentrations, as it*

*offers a unique opportunity to address aerosol data gaps in Asia. Moreover, South Korea has nationwide air quality monitoring stations, allowing us to obtain continuous and large data samples (PM10 and PM2.5) for validating the satellite data. To better utilize GEMS AOD for ground-level PM estimation, we employed machine learning models, which offer the advantage of experimenting with a wide range of input variables. For example, ground-level aerosols are influenced not only by meteorological conditions but also by precursor pollutants such as $SO_2$ and $NO_2$. Machine learning allows for the efficient integration and processing of these diverse datasets, enhancing the ability to utilize AOD for aerosol estimation.*

*[...] We aim to employ practical ML models to assess the potential of GEMS AOD data for sub-daily PM estimation, without prioritizing to achieve the best possible model performance. More advanced ML models or different modelling approaches (e.g. chemical transport models) could improve performance, but this is beyond the scope of our study. Furthermore, to our knowledge, this is the first evaluation of GEMS AOD applications, providing baseline results for future model comparisons and development. This approach also applies to input selection. While we consider a relatively wide range of input variables, including both meteorological and chemical data, additional variables can be tested, and model performance can be compared. In this context, ML models are particularly advantageous, as they can incorporate a broad spectrum of variables, including those typically not used in process-based models. However, ML performance is highly dependent on the quality of the training data; therefore, careful attention to data quality is essential.*

2. The introduction of the manuscript should include a brief description on the sections of the manuscript. Results and discussion should be a separate section from data and methodology. I suggest separating data and methodology as separate sections, as this manuscript lacks proper description on the methodology (there is too little information on the machine learning method (RF), selection criteria for input variables, ranges of the input variables.

⇨ *Thank you for the suggestion. First, we have added the description on the sections in the end of the introduction: In the following sections, we first describe the data and its preprocessing in Section 2. Section 3 details the methodology, including the machine learning models employed for PM estimation. In Section 4, we present the results and discuss their implications, followed by conclusions and future research directions in Section 5.*

⇨ *Second, we have separated the data and methods, and we have added more description on the model in Methods: RF operates by constructing multiple decision trees during training and aggregating their predictions to enhance accuracy and avoid overfitting. It is widely recognized for its ability to efficiently handle non-linear relationships in data and is often used for estimating PM concentrations. We also use XGBoost, which is similarly based on decision trees. However, XGBoost builds trees sequentially, allowing each tree to learn from the errors of the previous one, and is generally considered to outperform RF.*

⇨ *Lastly, we have also added more description on the input variables: Those input variables are selected with reference to previous studies (Yang et al., 2020; Handschuh et al. 2022), including Seo et al. (2015), which conducted experiments in South Korea. [...] We obtain input data from reanalysis datasets, which are readily available across all*

*areas within the GEMS satellite observation coverage. This ensures that the experiment conducted in this study can be easily extended to other locations, including other Asian countries, particularly in areas where meteorological measurements are unavailable. Additionally, reanalysis datasets provide consistent and reliable data updates over space and time. Nonetheless, it is well known that gases such as CO, $NO_2$, and $SO_2$ can influence PM formation mechanisms either directly or indirectly. Therefore, we also incorporate chemical data measured at the AirKorea stations as additional input features. In this way, we can evaluate the potential improvements in PM estimation using AOD when supplemented with additional information, and we report the corresponding results.*

3.  What is the sample size of the data used in RF?

⇨  *Since the models are applied to individual station points, the size of the training data varies for each model (station). To illustrate this, we have added the following figure as a Supplementary Material, showing the distribution of training data sizes across all stations.*

[Figure]

4.  RF was selected to estimate PM concentrations out of some other ML methods. How do you evaluate the model effectiveness in this work? Model performance can also affect the conclusions you draw regarding the ability of GEMS AOD to accurately provide PM concentrations.

⇨  *The model effectiveness is evaluated through the k-fold cross validation, as explained in Methods;* *"The main analysis is based on model predictions obtained through five-fold cross-validation."* *Model correlations, linear regression slopes, and relative errors (which is newly added) are all based on the five-fold cross-validation. Please note that additionally we have employed another machine learning method, XGBoost, to compare the performance of different models and further validate our approach.*

5.  The first part of the results should be to validate the GEMS AOD retrievals

⇨  *We have reorganized the section to present the comparison between GEMS and AERONET AOD as the first subsection, with the following figure now labeled as Fig. 1.*

[Figure]

⇨ *The added text is as follows: First, we directly compare the GEMS AOD data with ground-based AOD measurements from the AERONET. As shown in Fig.1a, the temporal variations of AODs at each station exhibit overall good correlations (Pearson's r), ranging from 0.68 to 0.89. When the entire time series from all AERONET sites are compared, the correlation remains strong (r = 0.77), although GEMS tends to underestimate AOD compared to the ground-based AOD measurements, as indicated by the linear regression slope (0.66) in Fig.1b. Furthermore, Fig. 1c demonstrates that this underestimation is consistent across most AERONET AOD ranges, with overestimation can also occur at very low AOD values. A study on the early version of GEMS L2 algorithm prior to the launch of GEMS also reported high correlation but slight underestimation of GEMS AOD relative to AERONET (Kim et al., 2020). Recent studies using GEMS L2 data in Asia regions have reported similar findings (e.g. Cho et al. 2024; Jang et al. 2024).*

6. The labeling of PM measurements used in RF, and the PM estimations, is vague. Make it more distinct.

⇨ *We have reviewed and revised the labeling throughout the paper to ensure a clear distinction between PM measurements and PM estimations.*

7. The use of mean vs error plots would be a better way of understanding the model performance rather than comparing the correlation coefficients. (Refer, Bland-Altman analysis)

⇨ *Thank you for the suggestion. We have conducted additional analysis on the model error structure using relative errors, defined as the difference between estimated and measured PM divided by the measured values, at each 10 percentile of the measured PM. The result is now reported in Fig. 3, as shown below.*

[Figure]

**Fig. 3 Performance of RF models in estimating ground-level PM concentrations.** *(a) Density scatter plot between measured PM10 and GEMS AOD across all stations. (b) Density scatter plot between measured PM10 and model-estimated PM10. The vertical and horizontal lines represent the corresponding median values. The thick solid line is the regression line, and the dotted diagonal line is the one-to-one. (c) Relative errors, defined as the difference between estimated and measured PM divided by the measured values, are shown at each 10th percentile of measured PM10. Panels (d), (e), and (f) correspond to (a), (b), and (c), respectively, but for PM25.*

8. Add more details description on SHAP analysis.

⇨ *We have added the following sentences in Methods. However, the primary disadvantage of machine learning is its 'black-box' nature, meaning we cannot fully understand why it produces certain estimations. To address this limitation and examine the role of the input features, we further use SHapley Additive exPlanations (SHAP) and quantify the relative importance of the considered input features on the model's predictions. SHAP is an explainable machine learning method based on Shapley values, which measure the marginal contribution of each predictor to the model's output or prediction across all the possible predictor combinations.*

⇨ *We have added more detailed descriptions of the SHAP analysis results, as follows: Furthermore, we use SHAP to examine the relative importance of the considered input features on the model's estimations (see Methods). As SHAP is computed for individual observations, we take the mean of absolute SHAP values for each input variable across all the estimations to explain its global feature contributions (Fig. 4). For both PM10 and PM25 estimations, AOD has a greater influence on model performance than most meteorological variables, as expected from its relatively strong correlation with ground aerosols (Fig. 2a), demonstrating the effectiveness of satellite information for ground aerosol simulations. [...] Among other meteorological variables, relative humidity (RH) emerges as one of the most important predictors for PM10 estimations. RH influences wet deposition of PM and also characterises seasonal variations in PM concentrations. For instance, in Korea, PM10 tends to increase due to wildfire emissions during dry seasons or from relatively coarse particles transported by Asian dust in the spring season. On the other hand, BLH has a greater importance for PM25. As aerosols are*

*primarily confined to the planetary boundary layer, BLH is a good proxy for estimating the height of the aerosol layer (Lee et al., 2024) and can help relate columnar satellite data to surface aerosol values (Handschuh et al., 2022; Gupta and Christopher, 2009a). The stronger importance of BLH for PM25 suggests its effectiveness in capturing the vertical distribution of finer aerosols. Previous studies have suggested a strong relationship between BLH and PM25 due to well-recognized positive feedbacks (e.g. Li et al., 2017; Su et al., 2022), which further underscores the importance of BLH. The significant influence of both BLH and RH on the relationship between AOD and PM has also been reported by Zheng et al., 2017.*

9. L 59-61 Include more details about GEMS instrument (uncertainties, wavelength channels). Do you perform any pixel averaging?

⇨ *We have added the following explanation about the GEMS instrument: The GEMS measures radiance in the 300–500 nm range with a spectral resolution of 0.6 nm and retrieves aerosol properties. The GEMS aerosol retrieval algorithm (AERAOD) uses the optimal estimation (OE) method, which integrates satellite-observed radiances with initial estimates of aerosol properties, including AOD, derived using the two-channel inversion approach employed by the OMAERUV algorithm (Torres et al., 2007).*

⇨ *No, we do not perform any pixel averaging. GEMS Level 2 data is provided on a pixel basis, and we select the pixel closest to the ground station at each time step, resulting in average distances ranging from 1.5 to 2.7 km. While gridding or interpolation could be applied to align the spatial resolution of GEMS with ground station data, such preprocessing may introduce additional uncertainties. This is especially relevant given that GEMS data contain many spatial gaps due to cloud contamination and other factors.*

⇨ *We discuss the potential uncertainty from the spatial mismatch of the data in the main text as follows: While GEMS data could be collocated to the exact location of ground stations through interpolation, additional preprocessing may introduce further uncertainties. Furthermore, we confirm that the spatial distances between GEMS pixels and ground stations do not significantly affect PM estimation performance (see Fig. S2).*

[Figure]

***Figure S2.*** *PM estimation performance by distance between data sources and measurement stations. (a) and (b) show the distribution of distances between GEMS AOD observation pixels and PM measurement stations, along with*

*average model performance (correlations) for PM10 and PM2.5, respectively. (c) and (d) depict the same, but for the distances between meteorological input data (i.e. ERA5-Land) and PM measurement stations.*

10. You need to add a description on GEMS AOD retrieval algorithm and explain possible uncertainties in AOD retrievals. The citation is not enough.

⇨ *Thank you for pointing this out. We have added a more explanation of the GEMS AOD retrieval algorithm (please see our response to Comment #9). Additionally, we have expanded on the results of the comparison between GEMS and AERONET AOD, as addressed in our reply to Comment #5.*

⇨ *However, we would like to emphasize that the primary focus of our study is on the usefulness of GEMS AOD in estimating PM concentrations rather than a detailed analysis of the AOD retrieval process itself (see our response to Comment #1).*

⇨ *Nonetheless, to address the Reviewer's concern, we have included a discussion of potential sources of bias in GEMS AOD retrievals, drawing from recent studies; Cho et al. (2024) specifically compared GEMS and AERONET AOD measurements in Asia and pointed out that the absence of region-specific aerosol type information in the GEMS aerosol model, as well as inaccuracies in cloud-masking processes, may negatively impact the accuracy of GEMS AOD data.*

11. L 74 – Do you perform any geolocation of data? How do you collocate reanalysis data? What is the maximum possible difference in the colocation of ground-based PM concentrations, AOD and reanalysis data?

⇨ *We initially selected the closest AOD and reanalysis data to the ground-based PM measurements. However, we have updated the method as follows: Both gridded datasets are interpolated to the locations of AirKorea stations using inverse distance weighting (IDW) based on the four closest grid points. If data are missing in the nearest grid points (e.g., over ocean areas), the corresponding locations are excluded from the analysis.*

⇨ *Please note that the PM estimation results remain consistent after changing the geolocation method. For details on the distance differences between the datasets, please refer to our response and the corresponding figure in Comment #9.*

12. L 115 – Is this something evident across all AOD values? Are there any differences seen for PM estimations under lower AOD and higher AOD values. Is there any detection limit?

⇨ *Thank you for your question. As detailed in the figure provided in our response to Comment #7, we observe a relatively strong overestimation at low AOD values and a weaker underestimation at high AOD values. We did not explicitly set any detection limits. If such limits exist, they are assumed to be inherently learned by the machine learning model during the training process.*

13. L 154 – AERONET data should be introduced under the data section. How do you collocate AERONET data? What do you mean by closest? You should specify the

distance limit. How do you average temporal data. Does the difference between AERONET AOD and GEMS AOD lies with AEROENT AOD uncertainty?

⇨ *We have introduced the AERONET data in the data section and also explain how we prepared the data, as follows:* Finally, for direct comparison, we obtain ground-based AOD measurements from AERONET sites in South Korea. A total of nine stations are selected, where data are available during the study period. AERONET provides highly accurate AOD measurements using Cimel Electronique Sun–sky radiometers, with an uncertainty of approximately 0.01-0.02. For this study, we use the version 3, level 2.0 quality-assured AOD at 440nm. For the comparison, GEMS AOD data within a 5 km radius of the AERONET sites are considered, and sub-hourly AERONET data are averaged within a temporal window of ±20 minutes around the GEMS observation time.

⇨ *Please note that, as shown in the figure provided in our response to Comment #5, the discrepancy exceeds AERONET AOD uncertainty. As explained in our response to Comment #10, this is likely due to additional factors, such as characteristics of the GEMS algorithm.*

14. L 63 – What is ARA?

⇨ *Aerosol Retrieval Algorithm. The text is revised.*

15. L 69 – What is ERA5?

⇨ *ECMWF Reanalysis v5 (ERA5) data. The text is revised.*

16. L 75 – How did you perform the AOD-PM simulations? Or do you mean estimations?

⇨ *We meant the PM estimation from the model simulations. The text is revised.*

17. L 83 – 89 This paragraph should go under the data section

⇨ *Thank you for the suggestion. We have moved the part of the paragraph to the data section.*

18. L 157 – Has it been observed for low AOD or high AOD?

⇨ *The underestimation is strong in the high AOD, but observed across all ranges. Please refer to the figure in our response to Comment #5,*

19. L 164 -165 Does the GEMS AOD algorithm consider any non-sphericity dust? You should add a description about the AOD algorithm

⇨ *The current GEMS aerosol retrieval algorithm primarily assumes spherical particles in its calculations. We have added a more detailed description of the algorithm and included references to a relevant recent study (see our response to Comment #6).*

20. L 176 – Fig 6. What does n=1,2,… stand for?

⇨ *Thank you for pointing this out. It refers to the number of neighboring stations providing the training dataset. We have updated the figure caption to clarify this. Additionally, please note that we have slightly modified the experimental setup to demonstrate the potential of satellite-derived AOD and machine learning for estimating PM concentrations at ungauged locations.*

[Figure]

**Figure 7. Potential of satellite data in PM estimation at ungauged locations.** *(a) Correlation and (b) slope of the linear regression between the measured and estimated PM10 concentrations at each station. Data from the n closest neighboring sites are used to train the RF models, and the model performance is evaluated at the target station, where the training data is deliberately excluded for this experiment. The term 'gauged' indicates that the model is trained and tested at the same station (as shown in the main analysis in Fig. 2).*

---

## Author Response (AR1)

**Reply to Reviewer #1 for Manuscript of "Estimating hourly ground-level aerosols using GEMS aerosol optical depth: A machine learning approach" by O et al.**

This manuscript is purposed on the PM concentration estimation based on the GEMS AOD observation data.

Because the GEMS is the geostationary orbit satellite, the idea in this manuscript has advantage of diurnal change monitoring of PM concentration from the satellite measurement.

⇨ *We thank the reviewer for the thoughtful comments that have significantly contributed to improving the quality of our manuscript. Below, we provide point-by-point responses to each comment. Please note that the page and line numbers mentioned in our responses correspond to the marked-up version of the manuscript.*

⇨ *Here, we summarize the main revisions for the manuscript:*
  - ***Data Update:*** *The dataset has been extended to December 2023, covering two years.*
  - ***Additional Machine Learning Algorithm:*** *To enhance the robustness of our machine learning-based modeling, we have employed an additional algorithm, XGBoost (XGB).*
  - ***Expanded Input Features:*** *We have considered additional air pollutant data, such as $SO_2$, $NO_2$, and $O_3$, in the modeling process.*
  - ***The manuscript structure*** *is reorganized. Specifically:*
    - *Data and Methods are presented as separate sections.*
    - *A direct comparison of GEMS AOD and ground-measured AOD is reported in the main text to briefly validate the GEMS AOD retrievals.*
    - *Results with PM2.5 are reported in the main text.*

However, to understand and check this characteristics, this manuscript needs to some additional analysis. Especially, the machine learning method is not a perfect approach and its result can be changed by the input data selection.

For this reason, idealizing and analyzing the input variable for the machine learning method is essential to include the manuscript.

⇨ *We appreciate the reviewer's insightful comment. We agree that understanding the characteristics of the machine learning (ML) approach and its sensitivity to input data selection is crucial. To address this concern, we have conducted additional analyses focusing on the input data (Fig. 6). Nonetheless, we would like to clarify that our primary aim is not to develop the best model for estimating PM concentrations but rather to evaluate the potential of new satellite data (GEMS AOD) as a valuable input for PM estimation. In this context, ML serves as an optimal tool due to its flexibility and ease in testing new data alongside other inputs that may contribute to improving model performance.*

For the detail, I listed to the below.

1) Introduction: For the readability of the manuscript, the author will be added to the brief explanation of sections in the final part of the Introduction section.

⇨ *Thank you for the suggestion. We have added the description on the sections in the end of the introduction:* In the following sections, we first describe the data and its preprocessing in Sect. 2. Section 3 details the methodology, including the machine learning models employed for PM estimation. In Sect. 4, we present the results and discuss their implications, followed by conclusions and future research directions in Sect. 5." *at Lines 89–91.*

2) Adding the reference

- Because the study of satellite retrieved AOD was largely evaluated, the reference and related paragraph will be added before the paragraph of Line 33 (Before GEMS AOD study)

⇨ *We have added the following text in Introduction:* "[...] the accuracy of satellite AOD data needs to be validated to ensure their reliability for downstream applications, including ground-level PM estimation. Typically, this involves comparisons with ground-based measurements (e.g. Ogunjobi and Awoleye, 2019; Mangla et al., 2020). For instance, Choi et al. (2019) evaluated various satellite-derived AODs against ground-based measurements collected during the 2016 KORUS-AQ campaign in East Asia. Similarly, Cho et al. (2024) validated the performance of GEMS aerosol products against ground measured data. Both studies revealed a strong correlation between satellite and ground-based AOD measurements, demonstrating the utility of satellite-derived AOD for monitoring data-scarce regions." *at Lines 43–49.*

- L35: The references related to AOD definition and its retrieval method will be added, such as Go et al. (2020).

⇨ *Thank you for the suggestion. We have added more references including Torres et al. and (2007), Go et al. (2020):* "The GEMS aerosol retrieval algorithm (AERAOD) uses the optimal estimation (OE) method, which integrates satellite-observed radiances with initial estimates of aerosol properties, including AOD, derived from the two-channel inversion approach employed by the OMAERUV algorithm (Torres et al., 2007). The GEMS aerosol products provide final AOD at three wavelength channels with a nominal spatial resolution of 3.5 km x 8 km at Seoul. Further details about the GEMS aerosol retrievals can be found from NIER (2020) and Go et al. (2020)." *at Lines 104–109.*

- L56: What is 'Korea Environment Corporation'? Is that 'Korean Environmental Institute'?

⇨ *K-eco is a government-affiliated public institution under the Ministry of Environment in South Korea, and it is responsible for managing AirKorea. We have modified the text as* "[...] from the AirKorea real-time ambient air quality monitoring system operated by the Korea Environment Corporation, a government-affiliated public institution under the Ministry of Environment." *at Line 95.*

- L57: For the PM concentration measurement, the author will be added the related references.

⇨ *The following reference is added at Line 96: Hauck, H., Berner, A., Gomiscek, B., Stopper, S., Puxbaum, H., Kundi, M., and Preining, O.: On the equivalence of gravimetric PM data with TEOM and beta-attenuation measurements, J. Aerosol Sci., 35, 1135–1149, doi:10.1016/j.aerosci.2004.04.004, 2004.*

- L63: What is ARA? Need to clarify.

⇨ *Aerosol Retrieval Algorithm. We have modified the citation as "NIER (2020)" at Line 109.*

3) From the Section 2, this manuscript included both methodology and result parts. I suggest that this section separates the method section (e.g., Section 2) and Result section (e.g. Section 3), and the author will make sub-sections for the detailed explanation of each parts. In this version of the manuscript, method and result parts are too short to clarify the detailed machine learning method and the reason of variable selections. For this reason, the manuscript is not able to identify the difference of research compared to the several previous studies for PM estimation by the machine learning. The author have to include the table for the list of the selected variable and selection criteria.

⇨ *Following the reviewer's suggestion, we have restructured the manuscript for greater clarity. The manuscript is now organized into Section 2: Data, Section 3: Methods, and Section 4: Results. Furthermore, Section 4 has been divided into three subsections for better organization: Sect. 4.1 Evaluation of GEMS AOD retrievals, Sect. 4.2 Performance of PM estimation derived using GEMS AOD and machine learning models, and Sect. 4.3 Enhancing PM estimation and applications at ungauged locations. Please note that, in response to a suggestion from another reviewer, we now present Section 4.1 in the main text, whereas it was previously included in the Supplementary Material.*

⇨ *To address the reviewer's comment, we have explained the reason for the variable and data selections in Sect. 2, with a table listing the selected variables: "Those input variables are selected based on previous studies (Yang et al., 2020; Handschuh et al. 2022), including Seo et al. (2015), which examines the importance of incorporating meteorological data for accurate PM estimation in South Korea using satellite-derived AOD. [...] We obtain input data from reanalysis datasets, which are readily available across all areas within the GEMS satellite observation coverage. This ensures that the experiment conducted in this study can be easily extended to other locations, including other Asian countries, particularly in areas where meteorological measurements are unavailable. Additionally, reanalysis datasets provide consistent and reliable data updates over space and time. Nonetheless, it is well known that gases such as CO, NO2, and SO2 can influence PM formation mechanisms either directly or indirectly (Lee et al., 2024a). Therefore, we also incorporate chemical data measured at the AirKorea stations as additional input features. In this way, we can evaluate the potential improvements in PM estimation using AOD when supplemented with additional information, and we report the corresponding results. The input variables used in this study are listed in Table 1." at Lines 119-138. Please also see Table 1 on Page 20.*

4) L59-L64: For the data colocation between ground and satellite pixels, temporal colocation is clarfied in the manuscript. However, the spatial colocation between Airkorea and GEMS pixel, the manuscript is explained only the 'nearlest pixel'. How to be selected

the 'nearlest pixel'? Because the cloud contamination of the GEMS AOD value, some pixels have to be eliminated, and the spatial distance between two measurements will be far. Do you have criteria of the maximum distance? In addition, the ground observation stations are dense in the urban region, especially denser than the GEMS spatial resolution. In this case, the same GEMS pixel is duplicately selected in different AirKorea observation sites. In this case, how to correct the colocation method?

⇨ *Please note that GEMS L2 data is provided on a pixel basis, and we selected the 'nearest pixel,' not the 'nearest observation.' This means that GEMS AOD values at the nearest pixel are often missing due to issues like cloud contamination, leaving an average of 1,990 data pairs (~11%) per station over the two-year period.*

⇨ *The average distance between GEMS pixels and AirKorea stations ranges from 1.5 to 2.7 km, while the average distance among AirKorea stations is 7.2 km. Therefore, we believe the impact of pixel duplication is minimal.*

⇨ *Further, we have confirmed that the spatial distances between GEMS pixels and ground stations do not significantly affect PM estimation performance, as shown in the figure below. The figure is included in Supplementary.*

[Figure]

**Figure S7.** Model performance and data distribution as a function of the distance between input features and target variables. Panels (a) and (b) show results for GEMS AOD paired with PM10 and PM2.5, respectively, while panels (c) and (d) present results for ERA5 reanalysis meteorological variables paired with PM10 and PM2.5, respectively. Gray bars indicate the percentage of data within each distance range, and black dots connected by lines represent the average model performance (correlation) for each distance range.

- In addition, for the colocation between observation and reanalysis dataset, what kind of interpolation method is used in this study? If you simply selected the 'nearest grid', it may affect the uncertainty.

⇨ *We agree with the Reviewer that simply selecting the "nearest grid" may introduce uncertainty. To address this, we have updated our colocation method to use Inverse distance weighting (IDW) is used. The updated method is explained as follows:* "Both gridded datasets are interpolated to the locations of AirKorea stations using inverse distance weighting based on the four closest grid points. If data are missing in the nearest grid points (e.g., over ocean areas), the corresponding locations are excluded from the analysis" *at Lines 127-129.*

5) L132: Boundary Layer Height (BLH) is not a linear relation to correct the PM concentration from satellite AOD. The BLH is roughly changing the PM concentration. But its sensitivity is also changed by the columnar concentration of aerosols. Did the author check the sensitivity change of BLH for the relationship between PM concentration and satellite AOD? (Including the reference survey)

⇨ *In our machine learning approach, the model learns patterns and relationships directly from the data without explicitly assuming linear or non-linear relationships like the sensitivity of BLH. While the analysis of the sensitivity change of BLH is beyond the scope of our study, we acknowledge the importance of understanding these physical relationships.*

⇨ *To address this, we have expanded our explanation of the role of BLH as follows: "BLH is a good proxy with which to estimate the height of the aerosol layer and can help relate columnar satellite data to surface aerosol values. The stronger importance of BLH for PM2.5 estimation suggests its effectiveness in capturing the vertical distribution of finer aerosols. Previous studies have suggested a strong relationship between BLH and PM2.5 due to well-recognised positive feedbacks (e.g. Wang et al., 2019; Su et al., 2017), which further underscores the importance of BLH. The relationship between AOD and PM is highly sensitive to variations in BLH conditions, as noted in previous studies (Zheng et al., 2017). For instance, higher BLH facilitates greater vertical dispersion of aerosols, thereby reducing surface PM concentrations for a given AOD. While our machine learning approach inherently captures such complex interactions, future work could explore the explicit sensitivity of BLH with the AOD-PM relationship to improve physical interpretability." at Lines 276-283.*

6) Section 2.1: Although the supplement part include the PM2.5 result, the body of the manuscript is not shown the detailed analysis of PM2.5. I suggested that both the PM10 and PM2.5 estimation method and SHAP analysis will be included separately. Also, the detailed SHAP analysis results have to be included with the detailed analysis and explanations. If the author compares the difference between PM2.5 and PM10 estimation, it is possible to evaluate the contribution of aerosol types or absorptivity. In addition, for the explanation of PM10 concentration, the manuscript is confused about what is 'observed from AirKorea' and 'satellite retrieved PM concentrations'. The author will clarify the word for satellite-derived PM concentration and Ground-based observed PM concentration.

⇨ *We appreciate the reviewer's detailed feedback and suggestions. While it would be interesting to investigate aerosol types or absorptivity, SHAP values primarily indicate the importance of input features and do not provide direct evidence to analyze such detailed characteristics. Thus, we consider this aspect suitable for future research.*

⇨ *Following the reviewer's suggestion, we have included the results for PM25 in the main text, along with its SHAP analysis. Please note that a computational error in the SHAP analysis was identified and corrected, resulting in slight changes to the results. However, AOD remains one of the most important input predictors.*

⇨ *We have also expanded the explanation and discussion of the SHAP analysis as follows: "For both PM10 and PM2.5 estimations, AOD has a great influence on model performance, demonstrating the effectiveness of satellite information for ground aerosol simulations. The direct relationships between PM and meteorological variables are not easily characterised (see Fig. S4), but SHAP analysis indicates that temperature (TEMP)*

*is the most influential meteorological predictors for both PM10 and PM2.5 estimations. Similar results are observed for XGB (Fig S5). [...]*

*Among other meteorological variables, relative humidity (RH) emerges as one of the most important predictors, particularly for PM10 estimations. RH influences wet deposition of PM and also characterises seasonal variations in PM concentrations. For instance, in Korea, PM10 tends to increase due to wildfire emission during dry seasons or from relatively coarse particles transported by Asian dust in the spring season. On the other hand, boundary layer height (BLH) has a greater importance for PM2.5 estimations. Given that aerosols are primarily confined to the planetary boundary layer, BLH is a good proxy for estimating the height of the aerosol layer (Lee et al., 2024) and can help relate columnar satellite data to surface aerosol values (Handschuh et al., 2022; Gupta and Christopher, 2009a). The stronger importance of BLH for PM2.5 estimation suggests its effectiveness in capturing the vertical distribution of finer aerosols. Previous studies have suggested a strong relationship between BLH and PM2.5 due to well-recognised positive feedbacks (e.g. Wang et al., 2019; Su et al., 2017), which further underscores the importance of BLH. [..]" at Lines 257-279.*

⇨ *To address the confusion regarding terminology, we have carefully reviewed and revised the manuscript to clearly distinguish between satellite-derived and ground-based observed PM concentrations.*

7) Figure 4 and 5: Re-arrange the time scale (24 hours -> Daytime)

⇨ *We have revised Figure 4 to include daytime data only, as suggested. It is now labeled as Fig. 5 in the updated manuscript. Please note that the original Figure 5 has been removed in the revised version.*

8) L161-L170: The author mentioned that the main reason for estimated PM underestimation is due to the GEMS AOD underestimation. However, this study's method made the machine learning model based on the GEMS AOD. If so, the uncertainty characteristics of GEMS AOD is adopted in the machine learning modeling. Another possibility of the estimated PM underestimation is the false selection of variables or lack of the variable for the machine learning method. From several previous studies, the PM concentration is not affected only by the meteorological components, but also by the chemical processes. The author has to check the variable selections.

⇨ *We agree with the reviewer that machine learning models should account for biases inherent in GEMS AOD. Following the reviewer's suggestion, we conducted additional experiments by incorporating chemical data as input variables for PM simulations. These experiments demonstrated that including chemical data can indeed improve the model's performance. However, the primary aim of our study is to use 'commonly-available' input data, such as reanalysis meteorological data and GEMS AOD, to ensure the generalizability of our approach to other Asian countries, where ground measurements may not be readily available.*

⇨ *Therefore, we have reported the results of these additional experiments as a test analysis under Subsection 4.3. This highlights the potential for improved model performance when supplementary data, such as chemical pollutants from ground*

*stations, are incorporated. In fact, while model performance improves with the inclusion of additional input data, discrepancies between the measured and estimated PM concentrations remain. These discrepancies may stem from limitations in the quality of AOD data (e.g. biases compared to ground-based AOD measurements) and/or inherent limitations of the machine learning models. These issues warrant further investigation in future studies.*

⇨ *The new result is reported in Fig. 6. Please also refer to Sect. 4.3 on Page 10.*

[Figure]

**Figure 6. Performance improvement of RF models with the inclusion of pollutant data.** This figure is similar to Fig. 3, but shows results for models trained with additional locally available data ($O_3$, CO, $SO_2$, $NO_2$) from AirKorea stations. (a) Density scatter plot between measured PM10 and model-estimated PM10. (b) Relative errors are shown at each 10th percentile of measured PM10. Panels (c) and (d) correspond to (a) and (b), respectively, but for PM2.5.

9) For the Machine learning adaptation, do you have the criteria of minimum concentration of observed PM and minimum value of satellite retrieved AOD? Low concentration of aerosol cases may be affecting the overall performance of estimation.

⇨ *Inspired by the reviewer's comment, we conducted additional experiments using ML models trained exclusively on high PM concentration cases. Unfortunately, this approach did not result in any noticeable improvement in model performance (see the figure below). Instead, we found that model performance can be enhanced by incorporating more informative input data, such as chemical pollutant measurements (see our responses to the comment above).*

⇨ *While there may be other approaches to further improve ML model performance, we consider this to be outside the scope of the current study. Our primary objective is to evaluate the usefulness of GEMS AOD in PM estimation and provide a baseline model performance for future studies.*

[Figure]

[Figure]

10) L179-L184 and Figure 6: For the statistical score, a detailed explanation will be needed. In addition, in Figure 6, 'n0', 'n1',' n2', 'n4', and 'n8' are not explained in the caption of Figure 6 and the body of the manuscript. The author has to clarify the explanation of Figure 6 and the mean of the statistical score.

⇨ *We apologize for the lack of clarity in the explanation. It refers to the number of neighboring stations providing the training dataset. We have updated the figure caption to clarify this. Additionally, please note that we have slightly modified the experimental setup to demonstrate the potential of satellite-derived AOD and machine learning for estimating PM concentrations at ungauged locations.*

[Figure]

[Figure]

**Figure 7. Potential of satellite data in PM estimation at ungauged locations.** (a) Correlation and (b) mean relative error between the measured and estimated PM10 concentrations at each station. The RF models are trained using data from the $n$ closest neighboring sites, excluding the target station, where model performance is evaluated. The term 'gauged' indicates that the model is trained and tested at the same station (as shown in the main analysis in Fig. 2).

**Reply to Reviewer #2 for Manuscript of "Estimating hourly ground-level aerosols using GEMS aerosol optical depth: A machine learning approach" by O et al.**

The manuscript is based on the estimation of PM2.5 and PM10 from GEMS AOD. The main objective is to evaluate the effectiveness of GEMS AOD in estimating ground level PM concentrations. This study attempts to study how GEMS AOD can provide air quality estimates in a global scale, which is of great importance

⇨ *We thank the reviewer for their recognition of the value of our study and for the thoughtful comments that have significantly contributed to improving the quality of our manuscript. Below, we provide point-by-point responses to each comment. Please note that the page and line numbers mentioned in our responses correspond to the marked-up version of the manuscript.*

⇨ *Here, we summarize the main revisions for the manuscript:*
- *__Data Update:__ The dataset has been extended to December 2023, covering two years.*
- *__Additional Machine Learning Algorithm:__ To enhance the robustness of our machine learning-based modeling, we have employed an additional algorithm, XGBoost (XGB).*
- *__Expanded Input Features:__ We have considered additional air pollutant data, such as $SO_2$, $NO_2$, and $O_3$, in the modeling process.*
- *__The manuscript structure__ is reorganized. Specifically:*
  - *Data and Methods are presented as separate sections.*
  - *A direct comparison of GEMS AOD and ground-measured AOD is reported in the main text to briefly validate the GEMS AOD retrievals.*
  - *Results with PM2.5 are reported in the main text.*

However, there are few concerns regarding the formulation of the study and the structure of the manuscript. Given below are my suggestions.

1. The overall paper lacks adequate explanations and citations to corroborate the objective of the study and how it differs from existing studies/novelty. (Ex: are there any ML based studies for estimation of PM concentrations? What are the advantages of this method over the existing?)

⇨ *We appreciate the reviewer's comment and have clarified the study's objectives and novelty. Our research is one of the first to validate GEMS AOD data, focusing on its utility for PM estimation rather than direct comparisons with ground-based AOD. Please note that we do not aim to develop best models for specific regions; instead, we evaluate GEMS AOD's broader applicability, such as improving models with additional data (e.g., chemical data) or estimating PM in areas without ground observations (e.g., using neighboring stations). In this context, we use ML models that are flexible enough to integrate diverse input data and can be relatively easily implemented in other regions, unlike physics-based models that often require additional parameter calibrations.*

⇨ *These points have been added to the introduction and conclusions for better clarity. e.g.* *"Here, we first evaluate GEMS AOD data through a direct comparison with ground-based*

*AERONET observations over South Korea. However, we place greater emphasis on evaluating the utility of GEMS AOD for estimating ground-level PM concentrations, as it offers a unique opportunity to address aerosol data gaps in Asia (Wen et al., 2023). Moreover, South Korea has nationwide air quality monitoring stations, allowing us to obtain continuous and large data samples (PM10 and PM2.5) for validating the satellite data. To better utilise GEMS AOD for ground-level PM estimation, we employ machine learning models, which offer the advantage of experimenting with a wide range of input variables. For example, ground-level aerosols are not related to AOD only, but influenced by meteorological conditions or precursor pollutants such as sulfur dioxide and nitrogen dioxide. Machine learning allows for the efficient integration and processing of these diverse datasets, enhancing the ability to utilize AOD for aerosol estimation."* **at Lines 51-59**

*"[...] we aim to assess the potential of GEMS AOD data for sub-daily PM estimation using practical ML models, without prioritising of optimal model performance. While more advanced ML techniques or alternative modelling approaches (e.g. chemical transport models) could improve performance, they are beyond this work. Furthermore, to our knowledge, this is the first evaluation of GEMS AOD applications, providing baseline results for future model comparisons and development. This approach also applies to input selection. While we consider a relatively wide range of input variables, including both meteorological and chemical data, additional variables can be tested, and model performance can be compared. In this context, ML models are particularly advantageous, as they can incorporate a broad spectrum of variables, including those typically not used in process-based models. However, ML performance is highly dependent on the quality of the training data; therefore, careful attention to data quality is essential."* **at Lines 382-391**

2. The introduction of the manuscript should include a brief description on the sections of the manuscript. Results and discussion should be a separate section from data and methodology. I suggest separating data and methodology as separate sections, as this manuscript lacks proper description on the methodology (there is too little information on the machine learning method (RF), selection criteria for input variables, ranges of the input variables.

⇨ *Thank you for the suggestion. First, we have added the description on the sections in the end of the introduction:* *"In the following sections, we first describe the data and its preprocessing in Sect. 2. Section 3 details the methodology, including the machine learning models employed for PM estimation. In Sect. 4, we present the results and discuss their implications, followed by conclusions and future research directions in Sect. 5."* **at Lines 89-91**

⇨ *Second, we have separated the data and methods, and we have added more description on the models in 3. Methods:* *"RF operates by constructing multiple decision trees during training and aggregating their predictions to enhance accuracy and avoid overfitting. It is widely recognized for its ability to efficiently handle non-linear relationships in data and is often used for estimating PM concentrations (e.g. Hu et al., 2017; Guo et al. 2021). We also use XGBoost, which is similarly based on decision trees. However, XGBoost builds trees sequentially, allowing each tree to learn from the errors of the previous one, and is generally considered to outperform RF (Chen and Guestrin, 2016)."* **at Lines 150-154**

⇨ *Lastly, we have also added more description on the input variables: "Those input variables are selected based on previous studies (Yang et al., 2020; Handschuh et al. 2022), including Seo et al. (2015), which examines the importance of incorporating meteorological data for accurate PM estimation in South Korea using satellite-derived AOD. [...] We obtain input data from reanalysis datasets, which are readily available across all areas within the GEMS satellite observation coverage. This ensures that the experiment conducted in this study can be easily extended to other locations, including other Asian countries, particularly in areas where meteorological measurements are unavailable. Additionally, reanalysis datasets provide consistent and reliable data updates over space and time. Nonetheless, it is well known that gases such as CO, NO2, and SO2 can influence PM formation mechanisms either directly or indirectly (Lee et al., 2024a). Therefore, we also incorporate chemical data measured at the AirKorea stations as additional input features. In this way, we can evaluate the potential improvements in PM estimation using AOD when supplemented with additional information, and we report the corresponding results. The input variables used in this study are listed in Table 1" at Lines 119-138. Please also see Table 1 on Page 20.*

3. What is the sample size of the data used in RF?

⇨ *Since the models are applied to individual station points, the size of the training data varies for each model (station). To illustrate this, we have added the following figure as a Supplementary Material, showing the distribution of training data sizes across all stations, The corresponding text in the manuscript states: "GEMS data values are often missing at the closest pixel due to issues such as cloud contamination or sun glint, resulting in an average of 1,990 AOD-PM data pairs per station (see Fig. S1 in Supplementary)." at Lines 111-113.*

[Figure]

4. RF was selected to estimate PM concentrations out of some other ML methods. How do you evaluate the model effectiveness in this work? Model performance can also affect the conclusions you draw regarding the ability of GEMS AOD to accurately provide PM concentrations.

⇨ *The model effectiveness is evaluated through the k-fold cross validation, as explained in Methods; "The main analysis is based on model predictions obtained through five-fold cross-validation. [...] Consequently, while this approach provides robust estimates of model performance, the actual performance of a model trained on the entire dataset could be underestimated due to the reduced size of the training data during cross-validation." at Lines 158-162. Model correlations and relative errors are all based on the five-fold cross-validation. Please note that additionally we have employed another machine learning method, XGBoost, to compare the performance of different models and further validate our approach.*

5. The first part of the results should be to validate the GEMS AOD retrievals

⇨ *We have reorganized the section to present the comparison between GEMS and AERONET AOD as the first subsection, with the following figure now labeled as Fig. 1.*

[Figure]

**Figure 1. Comparison between GEMS AOD and AERONET AOD observations**. (a) Correlations between GEMS AOD (443 $nm$) and AERONET AOD (440 $nm$) at individual AERONET stations. (b) Density scatter plot between GEMS and AERONET AOD across all stations. The vertical and horizontal lines represent the corresponding median values. The thick solid line is the regression line, and the dotted diagonal line is the one-to-one. (c) Average relative errors, defined as the difference between GEMS AOD and AERONET AOD divided by AERONET AOD, are shown for different AERONET AOD ranges. These ranges are divided based on each 20th percentile of AERONET AOD. Shaded areas indicate the interquartile range of the errors within each range.

⇨ *The added text is as follows: "First, we directly compare the GEMS AOD data with ground-based AOD measurements from the AERONET (Giles et al., 2019). As shown in Fig. 1a, the temporal variations of AODs at each station exhibit overall good correlations, with Persons's r ranging from 0.68 to 0.89. When the entire time series from all AERONET sites are compared, the correlation remains strong (r-value = 0.77), although GEMS tends to underestimate AOD compared to the ground-based measurements, as indicated by the linear regression slope (slp=0.66) in Fig. 1b. Furthermore, Fig. 1c demonstrates that this underestimation is consistent across most AERONET AOD ranges, although overestimation can also occur at very low AOD values. A study on the early version of GEMS L2 algorithm prior to the launch of GEMS also reported high correlation but slight underestimation of GEMS AOD relative to AERONET (Kim et al., 2020). Recent studies using GEMS L2 data in Asia regions have reported similar findings (e.g. Cho et al., 2024; Jang et al., 2024)." at Lines 177-190.*

6. The labeling of PM measurements used in RF, and the PM estimations, is vague. Make it more distinct.

⇨ *We have reviewed and revised the labeling throughout the paper to ensure a clear distinction between PM measurements and PM estimations.*

7. The use of mean vs error plots would be a better way of understanding the model performance rather than comparing the correlation coefficients. (Refer, Bland-Altman analysis)

⇨ *We appreciate the Reviewer's comment. In response, we have conducted additional analysis on the model error structure using relative errors, defined as the difference between the estimated and measured PM values divided by the measured values. The results of this analysis are now presented in Fig. 3, as shown below.*

[Figure]

**Figure 3. Performance of RF models in estimating ground-level PM concentrations.** (a) Density scatter plot between measured PM10 and GEMS AOD across all stations. (b) Density scatter plot between measured PM10 and model-estimated PM10. The vertical and horizontal lines represent the corresponding median values. The thick solid line is the regression line, and the dotted diagonal line is the one-to-one. Both axes are displayed in log scale for better visualization. (c) Average relative errors, defined as the difference between estimated and measured PM divided by the measured values, are shown at predefined ranges using each decile of measured PM10. Shaded areas indicate the 25th-75th percentiles within each range. The x-axis is in log scale. Panels (d), (e), and (f) correspond to (a), (b), and (c), respectively, but for PM2.5.

⇨ *During this additional analysis, we realised that the previous scatter plot did not clearly illustrate the data baises To address this, we now present the same results but using log-scaled axes. Consequently, this adjustment allows us to observe more detailed patterns in the direction of errors (underestimation vs. overestimation), and we have reported these findings in the manuscript at Lines 228-250.*

8. Add more details description on SHAP analysis.

⇨ *We have added the following sentences in Methods.* "However, the primary disadvantage of machine learning is its 'black-box' nature, meaning we cannot fully understand why it produces certain estimations. To address this limitation and examine the role of the input features, we further use SHapley Additive exPlanations (SHAP) and

*quantify the relative importance of the considered input features on the model's predictions. SHAP is an explainable machine learning method based on Shapley values, which measure the marginal contribution of each predictor to the model's output or estimation, by evaluating how the model output changes when a feature is included or excluded across all possible feature combinations (Monar, 2018; Lundberg et al., 2020)" at Lines 166–172.*

⇨ *We have added more detailed descriptions of the SHAP analysis results, as follows: Furthermore, we use SHAP to examine the relative importance of the considered input features on the model's estimations (see Methods). As SHAP is computed for individual observations, we take the mean of absolute SHAP values for each input variable across all the estimations to explain its global feature contributions (Fig. 4). For both PM10 and PM25 estimations, AOD has a great influence on model performance, demonstrating the effectiveness of satellite information for ground aerosol simulations. [...]*

*Among other meteorological variables, relative humidity (RH) emerges as one of the most important predictors, particuraly for PM10 estimations. RH influences wet deposition of PM and also characterises seasonal variations in PM concentrations. For instance, in Korea, PM10 tends to increase due to wildfire emissions during dry seasons or from relatively coarse particles transported by Asian dust in the dry spring season. On the other hand, BLH has a greater importance for PM25 estimations. As aerosols are primarily confined to the planetary boundary layer, BLH is a good proxy for estimating the height of the aerosol layer (Lee et al., 2024) and can help relate columnar satellite data to surface aerosol values (Handschuh et al., 2022; Gupta and Christopher, 2009a). The stronger importance of BLH for PM25 suggests its effectiveness in capturing the vertical distribution of finer aerosols. Previous studies have suggested a strong relationship between BLH and PM2.5 due to well-recognised positive feedbacks (e.g. Su et al., 2017; Wang et al., 2019), which further underscores the importance of BLH. The relationship between AOD and PM is highly sensitive to variations in BLH conditions, as noted in previous studies (Zheng et al., 2017). For instance, higher BLH facilitates greater vertical dispersion of aerosols, thereby reducing surface PM concentrations for a given AOD. While our machine learning approach inherently captures such complex interactions, future work could explore the explicit sensitivity of BLH with the AOD-PM relationship to improve physical interpretability." at Lines 252–283.*

9. L 59-61 Include more details about GEMS instrument (uncertainties, wavelength channels). Do you perform any pixel averaging?

⇨ *We have added the following explanation about the GEMS instrument: "[...] The GEMS measures radiance in the 300–500 nm range with a spectral resolution of 0.6 nm and retrieves aerosol properties. The GEMS aerosol retrieval algorithm (AERAOD) uses the optimal estimation (OE) method, which integrates satellite-observed radiances with initial estimates of aerosol properties, including AOD, derived from the two-channel inversion approach employed by the OMAERUV algorithm (Torres et al., 2007). [...]" at Lines 103–106.*

⇨ *No, we do not perform any pixel averaging. GEMS data is provided on a pixel basis, and we select the pixel closest to the ground station at each time step, resulting in average distances ranging from 1.5 to 2.7 km. While gridding or interpolation could be applied to align the spatial resolution of GEMS with ground station data, such preprocessing may introduce additional uncertainties. This is especially relevant given that GEMS data contain many spatial gaps due to cloud contamination and other factors.*

⇨ *We discuss this point in Conclusions: "[..] no significant performance degradation due to this discrepancy is observed (Fig. S7), employing higher-resolution data or improved interpolation techniques for data processing could also be considered in the future." at Lines 394-396.*

[Figure]

**Figure S7.** Model performance and data distribution as a function of the distance between input features and target variables. Panels (a) and (b) show results for GEMS AOD paired with PM10 and PM2.5, respectively, while panels (c) and (d) present results for ERA5 reanalysis meteorological variables paired with PM10 and PM2.5, respectively. Gray bars indicate the percentage of data within each distance range, and black dots connected by lines represent the average model performance (correlation) for each distance range.

10. You need to add a description on GEMS AOD retrieval algorithm and explain possible uncertainties in AOD retrievals. The citation is not enough.

⇨ *Thank you for pointing this out. We have added a more explanation of the GEMS AOD retrieval algorithm (please see our response to Comment #9). Additionally, we have expanded on the results of the comparison between GEMS and AERONET AOD, as addressed in our reply to Comment #5.*

⇨ *However, we would like to emphasize that the primary focus of our study is on the usefulness of GEMS AOD in estimating PM concentrations rather than a detailed analysis of the AOD retrieval process itself (see our response to Comment #1).*

⇨ *Nonetheless, to address the Reviewer's concern, we have included a discussion of potential sources of bias in GEMS AOD retrievals, drawing from recent studies; "Cho et al. (2024) specifically compared GEMS and AERONET AOD measurements for the period from 2021 to 2022 in Asia and pointed out that the absence of region-specific aerosol type information in the GEMS aerosol model, along with inaccuracies in cloud-masking processes, may adversely impact the accuracy of GEMS AOD data." at Lines 195–198.*

11. L 74 – Do you perform any geolocation of data? How do you collocate reanalysis data? What is the maximum possible difference in the colocation of ground-based PM concentrations, AOD and reanalysis data?

⇨ *We initially selected the closest AOD and reanalysis data to the ground-based PM measurements. However, we have updated the method as follows: "Both gridded datasets are interpolated to the locations of AirKorea stations using inverse distance weighting based on the four closest grid points. If data are missing in the nearest grid points (e.g., over ocean areas), the corresponding locations are excluded from the analysis" at Lines 127–129.*

⇨ *Please note that the PM estimation results do not change significantly after changing the geolocation method. For details on the distance differences between the datasets, please refer to our response and the corresponding figure in Comment #9.*

12. L 115 – Is this something evident across all AOD values? Are there any differences seen for PM estimations under lower AOD and higher AOD values. Is there any detection limit?

⇨ *Thank you for your question. As detailed in the figure provided in our response to Comment #7 (Fig. 3 in the manuscript), we observe an overestimation at low AOD values and a relatively strong underestimation at high values. We did not explicitly set any detection limits. If such limits exist, they are assumed to be inherently learned by the machine learning model during the training process.*

13. L 154 – AERONET data should be introduced under the data section. How do you collocate AERONET data? What do you mean by closest? You should specify the distance limit. How do you average temporal data. Does the difference between AERONET AOD and GEMS AOD lies with AEROENT AOD uncertainty?

⇨ *Following the Revierw's suggestion, we have introduced AERONET data in Sect. 2 Data: "Finally, for direct comparison, we obtain ground-based AOD measurements from AERONET sites in South Korea. A total of nine stations are selected, where data are available during the study period. AERONET provides highly accurate AOD measurements using Cimel Electronique Sun–sky radiometers, with an uncertainty of approximately 0.01-0.02 (Eck et al., 2019; Giles et al., 2019). For this study, we use the version 3, level 2.0 quality-assured AOD at 440 nm. For the comparison, GEMS AOD data within a 5 km radius of the AERONET sites are considered, and sub-hourly AERONET data are averaged within a temporal window of ±20 minutes around the GEMS observation time." at Lines 140–145.*

⇨ *Please note that, as shown in the figure provided in our response to Comment #5, the discrepancy exceeds AERONET AOD uncertainty. This is likely attributable to additional factors, such as the specific characteristics of the GEMS algorithm. This point is discussed in the manuscript at Lines 192–200.*

14. L 63 – What is ARA?

⇨ *Aerosol Retrieval Algorithm. Corrected as "NIER (2020)" at line 109.*

15. L 69 – What is ERA5?

⇨ *ECMWF Reanalysis v5 (ERA5) data. The text is revised.*

16. L 75 – How did you perform the AOD-PM simulations? Or do you mean estimations?

⇨ *We meant the PM estimation from the model simulations. The text is revised.*

17. L 83 – 89 This paragraph should go under the data section

⇨ *Thank you for the suggestion. We have moved the paragraph to the data section.*

18. L 157 – Has it been observed for low AOD or high AOD?

⇨ *The underestimation is slightly stronger for high AOD, but observed across all ranges. Please refer to the figure in our response to Comment #5,*

19. L 164 -165 Does the GEMS AOD algorithm consider any non-sphericity dust? You should add a description about the AOD algorithm

⇨ *The current GEMS aerosol retrieval algorithm primarily assumes spherical particles in its calculations. We have removed the corresponding text. We have added a more detailed description of the algorithm and included references to a relevant recent study (see our response to Comments #9 and #10).*

20. L 176 – Fig 6. What does n=1,2,… stand for?

⇨ *Thank you for pointing this out. It refers to the number of neighboring stations providing the training dataset. We have updated the figure caption to clarify this. Additionally, please note that we have slightly modified the experimental setup to demonstrate the potential of satellite-derived AOD and machine learning for estimating PM concentrations at ungauged locations.*

[Figure]

[Figure]

**Figure 7. Potential of satellite data in PM estimation at ungauged locations.** (a) Correlation and (b) mean relative error between the measured and estimated PM10 concentrations at each station. The RF models are trained using data from the $n$ closest neighboring sites, excluding the target station, where model performance is evaluated. The term 'gauged' indicates that the model is trained and tested at the same station (as shown in the main analysis in Fig. 2).